# GRADIENT MATCHING FOR DOMAIN GENERALIZATION

**Yuge Shi**[*]
University of Oxford
yshi@robots.ox.ac.uk

**Jeffrey Seely**
Meta Reality Labs
jseely@fb.com

**Philip H.S. Torr**
University of Oxford
philip.torr@eng.ox.ac.uk

**N. Siddharth**
The University of Edinburgh
& The Alan Turing Institute
n.siddharth@ed.ac.uk

**Awni Hannun**[†]
Facebook AI Research
awni@fb.com

**Nicolas Usunier**
Facebook AI Research
usunier@fb.com

**Gabriel Synnaeve**
Facebook AI Research
gab@fb.com

## ABSTRACT

Machine learning systems typically assume that the distributions of training and test sets match closely. However, a critical requirement of such systems in the real world is their ability to generalize to unseen domains. Here, we propose an *inter-domain gradient matching* objective that targets domain generalization by maximizing the inner product between gradients from different domains. Since direct optimization of the gradient inner product can be computationally prohibitive — it requires computation of second-order derivatives —- we derive a simpler first-order algorithm named Fish that approximates its optimization. We perform experiments on the WILDS benchmark, which captures distribution shift in the real world, as well as the DOMAINBED benchmark that focuses more on synthetic-to-real transfer. Our method produces competitive results on both benchmarks, demonstrating its effectiveness across a wide range of domain generalization tasks. Code is available at https://github.com/YugeTen/fish.

## 1 INTRODUCTION

The goal of domain generalization is to train models that performs well on unseen, out-of-distribution data, which is crucial in practice for model deployment in the wild. This seemingly difficult task is made possible by the presence of multiple distributions/domains at train time. As we have seen in past work (Arjovsky et al., 2019; Gulrajani and Lopez-Paz, 2020; Ganin et al., 2016), a key aspect of domain generalization is to learn from features that remain *invariant* across multiple domains, while ignoring those that are *spuriously correlated* to label information (as defined in Torralba and Efros (2011); Stock and Cisse (2017)). Consider, for example, a model that is built to distinguish between cows and camels using

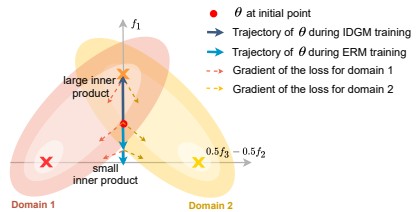

Figure 1: Isometric projection of training with ERM (blue) vs. our objective (dark blue), using data from Figure 2.

photos collected in nature under different climates. Since CNNs are known to have a bias towards texture (Geirhos et al., 2018; Brendel and Bethge, 2019), if we simply try to minimize the average loss across different domains, the classifier is prone to spuriously correlate "cow" with grass and "camels" with desert, and predict the species using only the background. Such a classifier can be rendered useless when the animals are placed indoors or in a zoo. However, if the model could recognize that while the landscapes change with climate, the biological characteristics of the animals

---

[*]Work done during internship at Facebook AI Research.
[†]Now at Zoom.

(e.g. humps, neck lengths) remain invariant and use those features to determine the species, we have a much better chance at generalizing to unseen domains.

Similar intuitions have already motivated many approaches that consider learning "invariances" across domains as the main challenge of domain generalization. Typically, a lot of these work focus on learning invariant representations directly by removing the domain information (Ganin et al., 2016; Sun and Saenko, 2016; Li et al., 2018). In this work, we propose an inter-domain gradient matching (IDGM) objective. Instead of learning invariant features by matching the distributions of representations from different domains, our approach does so by encouraging consistent gradient directions across domains. Specifically, our IDGM objective augments the loss with an auxiliary term that maximizes the gradient inner product between domains, which encourages the alignment between the domain-specific gradients. By simultaneously minimizing the loss and matching the gradients, IDGM encourages the optimization paths to be the same for all domains, favouring invariant predictions. Figure 1 illustrates a motivating example described in Section 3.2: given 2 domains, each containing one invariant feature (orange cross) and one spurious feature (yellow and red cross). While empirical risk minimization (ERM) minimizes the average loss between these domains at the cost of learning spurious features only, IDGM aligns the gradient directions and is therefore able to focus on the invariant feature.

While the IDGM objective achieves the desirable learning dynamic in theory, naive optimization of the objective by gradient descent is computationally costly due to the second-order derivatives. Leveraging the theoretical analysis of Reptile, a meta-learning algorithm (Nichol et al., 2018), we propose to approximate the gradients of IDGM using a simple first-order algorithm, which we name Fish. Fish is simple to implement, computationally effective and as we show in our experiments, functionally similar to direct optimization of IDGM.

Our contribution is a simple but effective algorithm for domain generalization, which exhibits state-of-the-art performance on 13 datasets from recent domain generalization benchmark WILDS (Koh et al., 2020) and DOMAINBED (Gulrajani and Lopez-Paz, 2020). The strong performance of our method on a variety of datasets demonstrates that it is broadly applicable in different applications/subgenres of domain generalization tasks. We also perform detailed analysis in Section 4.4 to explain the effectiveness of our proposed algorithm.

## 2 RELATED WORK

**Domain Generalization**    In domain generalization, the training data is sampled from one or many source domains, while the test data is sampled from a new target domain. We will now discuss the five main families of approaches to domain generalization:

1. **Distributional Robustness (DRO)**: DRO approaches minimize the worst-case loss over a set of data distributions constructed from the training domains. Rojas-Carulla et al. (2015) proposed DRO to address *covariate shift* (Gretton et al., 2009a;b), where $P(Y|X)$ remains constant across domains but $P(X)$ changes. Later work also studied *subpopulation shift*, where the train and test distributions are mixtures of the same domains, but the mixture weights change between train and test (Hu et al., 2018; Sagawa et al., 2019);

2. **Domain-invariant representation learning**: This family of approaches to domain generalization aims at learning high-level features that make domains statistically indistinguishable. Prediction is then based on these features only. The principle is motivated by a generalization error bound for unsupervised domain adaptation (Ben-David et al., 2010; Ganin et al., 2016), but the approach readily applies to domain generalization (Gulrajani and Lopez-Paz, 2020; Koh et al., 2020). Algorithms include penalising the domain-predictive power of the model (Ganin et al., 2016; Wang et al., 2019; Huang et al., 2020), aligning domains through contrastive loss (Motiian et al., 2017), matching mean and variance of feature distributions across domains (Sun and Saenko, 2016), learning useful representations by solving Jigsaw puzzles (Carlucci et al., 2019), using the maximum mean discrepancy to match the feature distributions (Li et al., 2018b) or introducing training constraints across domains using mixup formulation (Yan et al., 2020).

3. **Invariant Risk Minimization (IRM)**: IRM is proposed by Arjovsky et al. (2019), which learns an intermediate representation such that the optimal classifiers (on top of this representation) of all domains are the same. The motivation is to exploit invariant causal effects between domains while

reducing the effect of domain-specific spurious correlations. From an optimization perspective, when IRM reaches its optimal, all the gradients (for the linear classifier) *has to be zero*. This is why IRM's solution won't deviate from ERM when ERM is optimal for every domain, which is not the case for our proposed IDGM objective due to the gradient inner product term.

4. **Data augmentation**: More recently, approaches that simulates unseen domains through specific types of data augmentation/normalization has been gaining traction. This includes work such as Zhou et al. (2020); Volpi and Murino (2019); Ilse et al. (2021), as well as Seo et al. (2019) which utilises ensemble learning.

5. **Gradient alignment**: Two concurrent work – Koyama and Yamaguchi (2021) and Parascandolo et al. (2021) – utilise similar gradient-alignment principle for domain generalization. Koyama and Yamaguchi (2021) proposes IGA, which learns invariant features by minimizing the *variance* of inter-domain gradients. The key difference between IGA and our objective is that IGA is completely identical to ERM when ERM is the optimal solution on every training domain, since the variances of the gradients will be zero. While they achieve the best performance on the training set, both IGA and ERM *could* in some cases, completely fail when generalizing to unseen domains (see Section 3.2 for such an example). Our method, on the contrary, biases towards non-ERM solutions as long as the gradients are aligned, and is therefore able to avoid this issue. Parascandolo et al. (2021) on the other hand, proposes to mask out the gradients that have opposite signs for different domains. Unlike their work that prunes gradients that are inconsistent, our approach actively encourage gradients from different domains to be consistent by maximizing the gradient inner product. Additionally, in Lopez-Paz and Ranzato (2017) we also see the application of gradient-alignment, however in this case it is applied under the continual learning setting to determine whether a gradient update will increase the loss of the previous tasks.

Apart from these algorithms that are tailored for domain generalization, a well-studied baseline in this area is ERM, which simply minimizes the average loss over training domains. Using vanilla ERM is theoretically unfounded (Hashimoto et al., 2018; Blodgett et al., 2016; Tatman, 2017) since ERM is guaranteed to work only when train and test distributions match. Nonetheless, recent benchmarks suggest that ERM obtains strong performance in practice, in many case surpassing domain generalization algorithms (Gulrajani and Lopez-Paz, 2020; Koh et al., 2020). Our goal is to fill this gap, using an algorithm significantly simpler than previous approaches.

**Connections to meta-learning**   There are close connections between meta-learning (Thrun and Pratt, 1998) and (multi-source) domain adaptation. In fact, there are a few works in domain generalization that are inspired by the meta-learning principles, such as Li et al. (2018a); Balaji et al. (2018); Li et al. (2019); Dou et al. (2019). Specifically, Li et al. (2020) also proposes to adapt Reptile for domain generalization tasks, however they study their method under the sequential learning setting, whereas our method can be trained on all domains and therefore learns faster, especially when the number of domains is large. In Ren et al. (2018), we also see the leveraging of gradient inner product in meta-learning, where it is used to determine the importance weight of training examples. We discuss the connection between our proposed algorithm to meta-learning in more details in Appendix A.1.

Note that our proposed algorithm Fish is similar to the Mean Teacher method (Tarvainen and Valpola, 2017), where a teacher model (equivalent to $\theta$ in Algorithm 1) is computed using a moving average of the student model (equivalent to $\tilde{\theta}$ in Algorithm 1).

## 3 METHODOLOGY

### 3.1 GOALS

Consider a training dataset $\mathcal{D}_{tr}$ consisting of $S$ domains $\mathcal{D}_{tr} = \{\mathcal{D}_1, \cdots, \mathcal{D}_S\}$, where each domain $s$ is characterized by a dataset $\mathcal{D}_s := \{(x_i^s, y_i^s)\}_{i=1}^{n_s}$ containing data drawn i.i.d. from some probability distribution. Also consider a test dataset $\mathcal{D}_{te}$ consisting of $T$ domains $\mathcal{D}_{te} = \{\mathcal{D}_{S+1}, \cdots, \mathcal{D}_{S+T}\}$, where $\mathcal{D}_{tr} \cap \mathcal{D}_{te} = \emptyset$. The goal of domain generalization is to train a model with weights $\theta$ that generalizes well on the test dataset $\mathcal{D}_{te}$ such that:

$$\underset{\theta}{\arg\min} \, \mathbb{E}_{\mathcal{D} \sim \mathcal{D}_{te}} \mathbb{E}_{(x,y) \sim \mathcal{D}} \left[ l((x,y); \theta) \right], \tag{1}$$



Figure 2: All 3 domains (rows) consist of 3 types of inputs (columns): 1) $x_1$, **left**: makes up for 50% of each domain, label is always 0, $x_1$ is always $[0, 0, 0, 0]$; 2) $x_2$, **middle**: makes up for 40% of each domain, label is always 1, $x_2$ changes for each domain; 3) $x_3$, **right**: makes up for 10% of each domain, labels are randomly assigned with 30% of $y = 1$ and 70% of $y = 0$, $x_3$ is always $[1, 0, 0, 0]$.

where $l((x, y); \theta)$ is the loss of model $\theta$ evaluated on $(x, y)$.

A naive approach is to apply ERM, which simply minimizes the average loss on $\mathcal{D}_{tr}$, ignoring the discrepancy between train and test domains:

$$\mathcal{L}_{\mathrm{erm}}(\mathcal{D}_{tr}; \theta) = \mathbb{E}_{\mathcal{D} \sim \mathcal{D}_{tr}} \mathbb{E}_{(x,y) \sim \mathcal{D}} \left[ l((x, y); \theta) \right]. \tag{2}$$

The ERM objective does not exploit the invariance across different domains in $\mathcal{D}_{tr}$ and could perform arbitrarily poorly on test data. We demonstrate this effect with the following simple linear example.

## 3.2 THE PITFALL OF ERM: A LINEAR EXAMPLE

Consider a binary classification setup where data $(x, y) \in \mathbb{B}^4 \times \mathbb{B}$, and a data instance is denoted $x = [f_1, f_2, f_3, f_4], y$. The train domains are $\{\mathcal{D}_1, \mathcal{D}_2\}$, and test domain is $\mathcal{D}_3$. The goal is to learn a linear model $Wx + b = y, W \in \mathbb{R}^4, b \in \mathbb{R}$ on the train data, such that the error on the test domain is minimized. The setup and dataset of this example is illustrated in Figure 2.

As we can see in Figure 2, $f_1$ is the *invariant feature* in this dataset, since the correlation between $f_1$ and $y$ is stable across different domains. The relationships between $y$ and $f_2, f_3$ and $f_4$ changes for $\mathcal{D}_1, \mathcal{D}_2, \mathcal{D}_3$, making them the *spurious features*. Importantly, if we consider one domain only, the spurious features $f_2, f_3$ and $f_4$ are a more accurate indicator of the label than the invariant feature $f_1$. For instance, using $f_2$ to predict $y$ can give 97% accuracy on $\mathcal{D}_1$, while using $f_1$ only achieves 93% accuracy.

Table 1: Performance comparison on the linear dataset.

| Method | train acc. | test acc. | $W$ | $b$ |
|--------|-----------|-----------|-----|-----|
| ERM | 97% | 57% | $[2.8, 3.3, 3.3, 0.0]$ | $-2.7$ |
| IDGM | 93% | 93% | $[0.4, 0.2, 0.2, 0.0]$ | $-0.4$ |
| Fish | 93% | 93% | $[0.4, 0.2, 0.2, 0.0]$ | $-0.4$ |

The performance of ERM on this simple example is shown in Table 1 (first row). From the trained parameters $W$ and $b$, we see that the model places most of its weights on spurious features $f_2$ and $f_3$. While this achieves the highest train accuracy (97%), the model cannot generalize to unseen domains and performs poorly on test accuracy (57%).

## 3.3 INTER-DOMAIN GRADIENT MATCHING (IDGM)

To mitigate the problem with ERM, we need an objective that learns from features that are invariant across domains. Let us consider the case where the train dataset consists of $S = 2$ domains $\mathcal{D}_{tr} = \{\mathcal{D}_1, \mathcal{D}_2\}$. Given model $\theta$ and loss function $l$, the expected gradients for data in the two domains is expressed as

$$G_1 = \mathbb{E}_{\mathcal{D}_1} \frac{\partial l((x, y); \theta)}{\partial \theta}, \quad G_2 = \mathbb{E}_{\mathcal{D}_2} \frac{\partial l((x, y); \theta)}{\partial \theta}. \tag{3}$$

The direction, and by extension, inner product of these gradients are of particular importance to our goal of learning invariant features. If $G_1$ and $G_2$ point in a similar direction, i.e. $G_1 \cdot G_2 > 0$, taking a gradient step along $G_1$ or $G_2$ improves the model's performance on both domains, indicating that

the features learned by either gradient step are invariant across $\{\mathcal{D}_1, \mathcal{D}_2\}$. This invariance cannot be guaranteed if $G_1$ and $G_2$ are pointing in opposite directions, i.e. $G_1 \cdot G_2 \leq 0$.

To exploit this observation, we propose to maximize the gradient inner product (GIP) to align the gradient direction across domains. The intended effect is to find weights such that the input-output correspondence is as close as possible across domains. We name our objective *inter-domain gradient matching* (IDGM), and it is formed by subtracting the inner product of gradients between domains $\widehat{G}$ from the original ERM objective. For the general case where $S \geq 2$, we can write

$$\mathcal{L}_{\text{idgm}} = \mathcal{L}_{\text{erm}}(\mathcal{D}_{tr}; \theta) - \gamma \underbrace{\frac{2}{S(S-1)} \sum_{i,j \in S}^{i \neq j} G_i \cdot G_j}_{\text{GIP, denote as } \widehat{G}}, \tag{4}$$

where $\gamma$ is the scaling term for $\widehat{G}$. Note that GIP can be computed in linear time as $\widehat{G} = ||\sum_i G_i||^2 - \sum_i ||G_i||^2$ (ignoring the constant factor). We can also compute the stochastic estimates of Equation (4) by replacing out the expectations over the entire dataset by minibatches.

We test this objective on our simple linear dataset, and report results in the second row of Table 1. Note that to avoid exploding gradient we use the normalized GIP during training. The model has lower training accuracy compared to ERM (93%), however its accuracy remains the same on the test set, much higher than ERM. The trained weights $W$ reveal that the model assigns the largest weight to the invariant feature $f_1$, which is desirable. The visualization in Figure 1 also confirms that by maximizing the gradient inner product, IDGM is able to focus on the feature that is common between domains, yielding better generalization performance than ERM.

## 3.4 OPTIMIZING IDGM WITH FISH

The proposed IDGM objective, although effective, requires computing the second-order derivative of the model's parameters due to the gradient inner product term, which can be computationally prohibitive. To mitigate this, we propose a first-order algorithm named Fish[1] that approximates the optimization of IDGM with inner-loop updates. In Algorithm 1 we present Fish. As a comparison, we also present direct optimization of IDGM using SGD in Algorithm 2.

---

**Algorithm 1** Fish.

1: **for** iterations $= 1, 2, \cdots$ **do**
2:     $\widetilde{\theta} \leftarrow \theta$
3:     **for** $\mathcal{D}_i \in \text{permute}(\{\mathcal{D}_1, \mathcal{D}_2, \cdots, \mathcal{D}_S\})$ **do**
4:         Sample batch $d_i \sim \mathcal{D}_i$
5:         $\widetilde{g}_i = \mathbb{E}_{d_i}\left[\dfrac{\partial l((x,y); \widetilde{\theta})}{\partial \widetilde{\theta}}\right]$  // Grad wrt $\widetilde{\theta}$
6:         Update $\widetilde{\theta} \leftarrow \widetilde{\theta} - \alpha\widetilde{g}_i$
7:     **end for**
8:     Update $\theta \leftarrow \theta + \epsilon(\widetilde{\theta} - \theta)$
9: **end for**

---

**Algorithm 2** Direct optimization of IDGM.

1: **for** iterations $= 1, 2, \cdots$ **do**
2:     $\widetilde{\theta} \leftarrow \theta$
3:     **for** $\mathcal{D}_i \in \text{permute}(\{\mathcal{D}_1, \mathcal{D}_2, \cdots, \mathcal{D}_S\})$ **do**
4:         Sample batch $d_i \sim \mathcal{D}_i$
5:         $g_i = \mathbb{E}_{d_i}\left[\dfrac{\partial l((x,y); \theta)}{\partial \theta}\right]$  // Grad wrt $\theta$
6:     **end for**
7:     $\bar{g} = \dfrac{1}{S}\sum_{s=1}^{S} g_s, \quad \widehat{g} = \overbrace{\dfrac{2}{S(S-1)}\sum_{i,j \in S}^{i \neq j} g_i \cdot g_j}^{\text{GIP (batch)}}$
8:     Update $\theta \leftarrow \theta - \epsilon\left(\bar{g} - \gamma(\partial\widehat{g}/\partial\theta)\right)$
9: **end for**

---

Fish performs $S$ inner-loop (*l3-l7*) update steps with learning rate $\alpha$ on a clone of the original model $\widetilde{\theta}$, and each update uses a minibatch $d_i$ from the domain selected in step $i$. Subsequently, $\theta$ is updated by a weighted difference between the cloned model and the original model $\epsilon(\widetilde{\theta} - \theta)$.

To see why Fish is an approximation to directly optimizing IDGM, we can perform Taylor-series expansion on its update in *l8*, Algorithm 1. Doing so reveals two leading terms: 1) $\bar{g}$: averaged gradients over inner-loop's minibatches (effectively the ERM gradient); 2) $\partial\widehat{g}/\partial\theta$: gradient of the minibatch version of GIP. Observing *l8* of Algorithm 2, we see that $\bar{g}$ and $\widehat{g}$ are actually the two gradient components used in direct optimization of IDGM. Therefore, Fish implicitly optimizes

---

[1]Following the convention of naming this style of algorithms after classes of vertebrates (animals with backbones).

IDGM by construction (up to a constant factor), avoiding the computation of second-order derivative $\partial \widehat{g}/\partial \theta$. We present this more formally for the full gradient $G$ in Theorem 3.1.

**Theorem 3.1** *Given twice-differentiable model with parameters $\theta$ and objective $l$. Let us define the following:*

$$G_f = \mathbb{E}[(\theta - \widetilde{\theta})] - \alpha S \cdot \bar{G}, \qquad \text{Fish update - } \alpha S \cdot \text{ERM grad}$$

$$G_g = -\partial \widehat{G}/\partial \theta, \qquad \text{grad of } \max_{\theta}(\widehat{G})$$

*where $\bar{G} = \frac{1}{S} \sum_{s=1}^{S} G_s$ and is the full gradient of ERM. Then we have*

$$\lim_{\alpha \to 0} \frac{G_f \cdot G_g}{\|G_f\| \cdot \|G_g\|} = 1.$$

Note that the expectation in $G_f$ is over the sampling of domains and minibatches. Theorem 3.1 indicates that when $\alpha$ is sufficiently small, if we remove the scaled ERM gradient component $\bar{G}$ from Fish's update, we are left with a term $G_f$ that is in similar direction to the gradient of maximizing the GIP term in IDGM, which was originally second-order. Note that this approximation comes at the cost of losing direct control over the GIP scaling $\gamma$ — we therefore also derived a smoothed version of Fish that recovers this scaling term, however we find that changing the value of $\gamma$ does not make much difference empirically. See Appendix B for more details.

The proof to Theorem 3.1 can be found in Appendix A. We follow the analysis from Nichol et al. (2018), which proposes Reptile for model-agnostic meta-learning (MAML), where the relationship between inner-loop update and maximization of gradient inner product was first highlighted. Nichol et al. (2018) found the GIP term in their algorithm to be over minibatches from the *same domain*, which promoted within-task generalization; in Fish we construct inner-loop using minibatches over *different domains* – it therefore instead encourages across-domain generalization. We compare the two algorithms in further details in Appendix A.1.

We also train Fish on our simple linear dataset, with results in Table 1, and see it performs similarly to IDGM – the model assigns the most weight to the invariant feature $f_1$, and achieves 93% accuracy on both train and test dataset.

## 4 EXPERIMENTS

### 4.1 CDSPRITES-N

**Dataset** We propose a simple shape-color dataset CDSPRITES-N based on the DSPRITES dataset (Matthey et al., 2017), which contains a collection of white 2D sprites of different shapes, scales, rotations and positions. CDSPRITES-N contains $N$ domains. The goal is to classify the shape of the sprites, and there is a shape-color deterministic matching that is specific per domain. This way we have shape as the invariant feature and color as the spurious feature. See Figure 3 for an illustration.

To construct the train split of CDSPRITES-N, we take a subset of DSPRITES that contains only 2 shapes (square and oval). We make $N$ replicas of this subset and assign 2 colors to each, with every color corresponding to one shape (e.g. yellow block in Figure 3a, pink $\rightarrow$ squares, purple $\rightarrow$ oval). For the test split, we create another replica of the DSPRITES-N subset, and randomly assign one of the $2N$ colors in the training set to each shape in the test set.

We design this dataset with CNN's texture bias in mind (Geirhos et al., 2018; Brendel and Bethge, 2019). If the value of $N$ is small enough, the model can simply memorize the $N$ colors that correspond to each shape, and make predictions solely based on colors, resulting in poor performance on the test set where color and shape are no longer correlated. Our dataset allows for precise control over the features that remains stable across domains and the features that change as domains change; we can also change the number of domains $N$ easily, making it possible to examine the effect $N$ has on the performance for domain generalization.

**Results** We train the same model using three different objectives including Fish, dicrect optimization of IDGM and ERM on this dataset with number of domains $N$ ranging from 5 to 50. Again, for direct optimization of IDGM, we use the normalized gradient inner product to avoid exploding gradient.

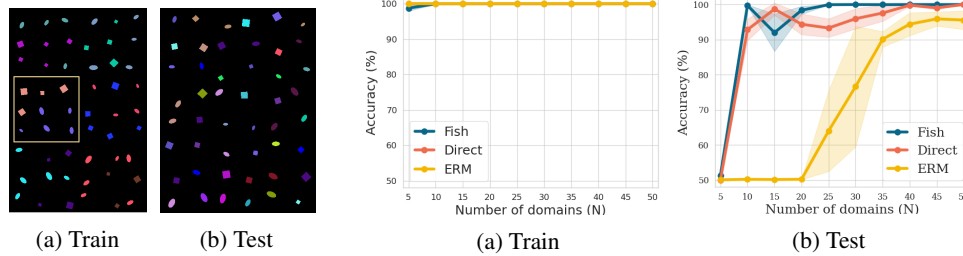

| (a) Train | (b) Test | (a) Train | (b) Test |

Figure 3: CDSPRITES-N visualization. Each 3x3 grid (e.g. yellow square) is one domain.

Figure 4: Performance of Fish, IDGM and ERM on CDSPRITES-N, with $N \in [5, 50]$

We plot the average train, test accuracy for each objective over 5 runs against the number of domains $N$ in Figure 4. We can see that the train accuracy is always $100\%$ for all methods regardless of $N$ (Figure 4a), while the test performance varies: Figure 4b shows that **direct** optimization of IDGM (red) and **Fish** (blue) obtain the best performances, with the test accuracy rising to over $90\%$ when $N \geq 10$ and near $100\%$ when $N \geq 20$. The predictions of **ERM** (yellow), on the other hand, remain nearly random on the test set up until $N = 20$, and reach $95\%$ accuracy only for $N \geq 40$.

This experiment confirms the following: 1) the proposed IDGM objective have much stronger domain generalization capabilities compared to ERM; 2) Fish is an effective approximation of IDGM, with similar performance to its direct optimization. We also plot the gradient inner product progression of Fish vs. ERM during training in Figure 9a, showing clearly that Fish does improve the gradient inner product across domain while ERM does not; 3) we also observe during training that Fish is about 10 times faster than directly optimizing IDGM, demonstrating its computational efficiency.

## 4.2 WILDS

**Datasets** We evaluate our model on the WILDS benchmark (Koh et al., 2020), which contains multiple datasets that capture real-world distribution shifts across a diverse range of modalities. We report experimental results on 6 challenging datasets in WILDS, and find Fish to outperform all baselines on most tasks. A summary of the WILDS datasets can be found in Appendix C. For hyperparameters including learning rate, batch size, choice of optimizer and model architecture, we follow the exact configuration as reported in the WILDS benchmark. Importantly, we also use the same model selection strategy used in WILDS to ensure a fair comparison. See details in Appendix D.

Table 2: Results on WILDS benchmark.

| | POVERTYMAP | CAMELYON17 | FMOW | CIVILCOMMENTS | iWILDCAM | AMAZON |
|---|---|---|---|---|---|---|
| | Worst-U/R Pearson r | Avg. acc. (%) | Worst acc. (%) | Worst acc. (%) | Macro F1 | 10-th per. acc. (%) |
| Fish | 0.30 ($\pm$1e-2) | 74.7 ($\pm$7.1) | 34.6 ($\pm$0.18) | 75.3 ($\pm$0.6) | 22.0 ($\pm$0.0) | 53.3 ($\pm$0.0) |
| IRM | 0.43 ($\pm$7e-2) | 64.2 ($\pm$8.1) | 30.0 ($\pm$1.37) | 66.3 ($\pm$2.1) | 15.1 ($\pm$4.9) | 52.4 ($\pm$0.8) |
| Coral | 0.44 ($\pm$6e-2) | 59.5 ($\pm$7.7) | 31.7 ($\pm$1.24) | 65.6 ($\pm$1.3) | 32.8 ($\pm$0.1) | 52.9 ($\pm$0.8) |
| Reweighted | - | - | - | 69.2 ($\pm$0.9) | - | 52.0 ($\pm$0.0) |
| GroupDRO | 0.39 ($\pm$6e-2) | 68.4 ($\pm$7.3) | 30.8 ($\pm$0.81) | 70.0 ($\pm$2.0) | 23.9 ($\pm$2.1) | 53.3 ($\pm$0.0) |
| ERM | 0.45 ($\pm$6e-2) | 70.3 ($\pm$6.4) | 32.3 ($\pm$1.25) | 56.0 ($\pm$3.6) | 31.0 ($\pm$1.3) | 53.8 ($\pm$0.8) |
| ERM (ours) | 0.29 ($\pm$1e-2) | 70.5 ($\pm$12.1) | 30.9 ($\pm$1.53) | 58.1 ($\pm$1.7) | 25.1 ($\pm$0.2) | 53.3 ($\pm$0.8) |

**Results** See a summary of results in Table 2, where we use the metrics recommended in WILDS for each dataset. Again, following practices in WILDS, all results are reported over 3 random seed runs, apart from CAMELYON17 which uses 10 random seeds and CIVILCOMMENTS which uses 5. We included additional results as well as a in-depth discussion on each dataset in Appendix C, and an ablation studies on Fish's hyperparameters in Appendix F and Appendix E. We make the following observations:

1. **Strong performance across datasets:** Considering results on all 6 datasets, Fish is the best performing algorithm on WILDS. It significantly outperforms baselines on 3 datasets and achieves similar level of performance to the best method on the other 3 (AMAZON and iWILDCAM). Fish's strong performance on different types of data and architectures such as RESNET (He et al., 2016), DENSENET (Huang et al., 2017) and DISTILBERT (Sanh et al., 2019) demonstrated it's capability to generalize to a diverse variety of tasks;

Table 3: Test accuracy (%) on DOMAINBED benchmark.

| | ERM | IRM | GroupDRO | Mixup | MLDG | Coral | MMD | DANN | CDANN | Fish (ours) |
|---|---|---|---|---|---|---|---|---|---|---|
| CMNIST | 51.5 (±0.1) | 52.0 (±0.1) | 52.1 (±0.0) | 52.1 (±0.2) | 51.5 (±0.1) | 51.5 (±0.1) | 51.5 (±0.2) | 51.5 (±0.3) | 51.7 (±0.1) | 51.6 (±0.1) |
| RMNIST | 98.0 (±0.0) | 97.7 (±0.1) | 98.0 (±0.0) | 98.0 (±0.1) | 97.9 (±0.0) | 98.0 (±0.1) | 97.9 (±0.0) | 97.8 (±0.1) | 97.9 (±0.1) | 98.0 (±0.0) |
| VLCS | 77.5 (±0.4) | 78.5 (±0.5) | 76.7 (±0.6) | 77.4 (±0.6) | 77.2 (±0.4) | 78.8 (±0.6) | 77.5 (±0.9) | 78.6 (±0.4) | 77.5 (±0.1) | 77.8 (±0.3) |
| PACS | 85.5 (±0.2) | 83.5 (±0.8) | 84.4 (±0.8) | 84.6 (±0.6) | 84.9 (±1.0) | 86.2 (±0.3) | 84.6 (±0.5) | 83.6 (±0.4) | 82.6 (±0.9) | 85.5 (±0.3) |
| OfficeHome | 66.5 (±0.3) | 64.3 (±2.2) | 66.0 (±0.7) | 68.1 (±0.3) | 66.8 (±0.6) | 68.7 (±0.3) | 66.3 (±0.1) | 65.9 (±0.6) | 65.8 (±1.3) | 68.6 (±0.4) |
| TerraInc | 46.1 (±1.8) | 47.6 (±0.8) | 43.2 (±1.1) | 47.9 (±0.8) | 47.7 (±0.9) | 47.6 (±1.0) | 42.2 (±1.6) | 46.7 (±0.5) | 45.8 (±1.6) | 45.1 (±1.3) |
| DomainNet | 40.9 (±0.1) | 33.9 (±2.8) | 33.3 (±0.2) | 39.2 (±0.1) | 41.2 (±0.1) | 41.5 (±0.1) | 23.4 (±9.5) | 38.3 (±0.1) | 38.3 (±0.3) | 42.7 (±0.2) |
| Average | 66.6 | 65.4 | 64.8 | 66.7 | 66.7 | 67.5 | 63.3 | 66.1 | 65.6 | 67.1 |

2. **Strong performance on different domain generalization tasks:** We make special note the CIVILCOMMENTS dataset captures *subpopulation shift* problems, where the domains in test are a subpopulation of the domains in train, while all other WILDS datasets depicts *pure domain generalization* problems, where the domains in train and test are disjointed. As a result, the baseline models for CIVILCOMMENTS selected by the WILDS benchmark are different from the methods used in all other datasets, and are tailored to avoiding systematic failure on data from minority subpopulations. We see that Fish works well in this setting too without any changes or special sampling strategies (used for baselines on CIVILCOMMENTS, see more in Table 10), demonstrating it's capability to perform in different domain generalization scenarios;

3. **Failure mode of domain generalization algorithms:** We noticed that on IWILDCAM and AMAZON, ERM is the best algorithm, outperforming all domain generalization algorithms except for Fish on AMAZON. We believe that these domain generalization algorithms failed due to the large number of domains in these two datasets — 324 for IWILDCAM and 7,676 for AMAZON. This is a common drawback of current domain generalization literature and is a direction worth exploring.

## 4.3   DOMAINBED

**Datasets** While WILDS is a challenging benchmark capturing realistic distribution shift, to test our model under the synthetic-to-real transfer setting and provide more comparisons to SOTA methods, we also performed experiments on the DOMAINBED benchmark (Gulrajani and Lopez-Paz, 2020). See a summary of DOMAINBED in Appendix H.

**Results** Following recommendations in DOMAINBED, we report results using training domain as validation set for model selection. See results in Table 3, reported over 5 random trials. Averaging the performance over all 7 datasets, Fish ranks second out of 10 domain generalization methods. It performs only marginally worse than Coral (0.1%), and is one of the three methods that performs better than ERM. This showcases Fish's effectiveness on domain generalization datasets with stronger focus to synthetic-to-real transfer, which again demonstrates its versatility and robustness on different domain generalization tasks.

## 4.4   ANALYSIS

We show extensively through empirical evaluation that Fish is very effective for a variety domain generalization tasks. In this section, we perform analysis to validate that Fish's strong performance is due to inter-domain gradient inner product maximization.

**Does Fish maximize gradient inner product (GIP) empirically?**
In Figure 5, we plot the progression of GIP during training using different objectives. We train both **Fish** (blue) and **ERM** (yellow) on CDSPRITES-N until convergence while tracking the normalized GIP between minibatches from different domains used in each inner-loop. To ensure a fair comparison, we use the exact same sequence of data for Fish and ERM (see Appendix I for more details).

From Figure 5, it is clear that during training, the normalized GIP of Fish increases, while that for ERM stays at the same value. The observations here shows that Fish is indeed effective in increasing/maintaining the level of inter-domain GIP.

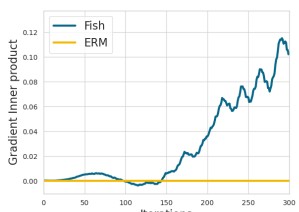

Figure 5: Gradient inner product values during the training for CDSPRITES-N (N=15).

Table 4: Test accuracy on four datasets, in three partitions are 1) two baselines Fish and ERM, 2) Fish with random grouping strategy (*Fish, RG*) and 3) ERM with domain grouping strategy (*ERM, DG, ∗*)

| | FMoW | VLCS | PACS | OfficeHome |
|---|---|---|---|---|
| Fish (ours) | 34.3 (±0.6) | 77.6 (±0.5) | 85.5 (±0.3) | 68.6 (±0.9) |
| ERM | 31.7 (±1.0) | 77.5 (±0.4) | 85.5 (±0.2) | 66.5 (±0.3) |
| Fish, RG | 33.4 (±1.7) | 77.7 (±0.3) | 83.9 (±0.7) | 66.5 (±1.0) |
| ERM, DG, prop. lr | 32.1 (±0.5) | 72.7 (±0.4) | 83.2 (±0.7) | 58.5 (±0.1) |
| ERM, DG, 0.1× prop. lr | 29.9 (±0.7) | 73.9 (±0.6) | 84.2 (±0.4) | 65.1 (±0.1) |
| ERM, DG, 0.01× prop. lr | 29.8 (±0.4) | 74.7 (±0.4) | 84.0 (±0.6) | 63.7 (±0.2) |

We conduct the same GIP tracking experiments for the WILDS datasets we studied as well to shed some lights on its efficiency — see Appendix G for results.

**Does maximizing GIP between random batches help domain generalization?** In this part of the analysis, we examine the effectiveness of another element of our algorithm – the construction of minibatches. We conduct experiments where data are grouped randomly instead of by domain for the inner-loop update. By doing so, we are still maximizing the inner product between minibatches, however not strictly between domain. We therefore expect the results to be slightly worse than Fish, and the bigger the domain gap is, the more advantage Fish has against the random grouping strategy.

We show the results for random grouping (*Fish, RG*) in Table 4. As expected, the random grouping strategy performs worse than Fish on all datasets. The experiment demonstrated that our algorithm does benefit from the domain grouping strategy, and that maximising GIP between random batches of data, while still achieving strong results, does not achieve the same domain generalization performance as Fish.

**Differences between Fish and ERM** We have shown through empirical evaluation that Fish and ERM are sufficiently different. Most notably, they converge to completely distinct minimum in our linear toy example and CDSPRITES experiment, resulting in large decrepencies in test accuracy.

However, some may notice that Fish as a first-order method shares algorithmic similarities to ERM. In fact, if we fix the meta learning rate of Fish $\epsilon = 1$ and use domain sampling for ERM, the two algorithms are equivalent. Although the optimal $\epsilon$s found through hyperparameter search are much smaller than 1, one might wonder if we can offset the increased meta learning rate $\epsilon$ by reducing the learning rate $\alpha$ proportionally, and achieve similar performance to Fish using simply ERM with domain grouping.

In Table 4 (*ERM, DG*), we verify that this is infeasible empirically. We train an ERM model with domain grouping strategy (equivalent to Fish with $\epsilon = 1$) using 3 different learning rates: 1) *prop. lr*, learning rate $\alpha$ is lowered proportionally to keep $\alpha\epsilon$ constant; 2) $0.1\times prop. lr$ and 3) $0.001\times prop. lr$ which further reduces the learning rate. We see that on all datasets, all three configurations result in worse performance than ERM and Fish. This shows that one cannot improve the results of ERM to match that of Fish by simply adjusting the learning rate/adopting domain grouping strategy, and that Fish's gain on domain generalization tasks is not due to its proximity to ERM.

In Appendix A.2, we also show that setting $\epsilon = 1$ is equivalent to setting $S$ as the total number of iterations during training, which causes the effect of maximizing GIP to diminish. This provide further support to our empirical finding in this analysis.

**Does maximising GIP help domain generalization?** In Parascandolo et al. (2021), authors show that by masking out the gradients that have opposite signs in different domains, they are able to trade-off some "learning speed" for prioritizing learning the invariances. Our principle for proposing the IDGM objective is identical, except in this case we actively *encourage* these gradients to be of consistent signs. This unfortunately, does result in computing the costly second-order derivative, which makes it difficult for us to empirically evaluate the IDGM objective on non-toy dataset[2]. We believe that the large performance gain of IDGM over ERM on the linear and CDSPRITES datasets, along with the detailed discussion on the relationship between gradient sign consistency and invariant representation, is enough to justify for IDGM's ability to improve domain generalization.

---

[2]For context, training an IDGM model on VLCS dataset with batch size 32 and standard RESNET18 backbone does not fit a NVIDIA V100 (32GB) GPU.

## 5 CONCLUSION

In this paper we presented inter-domain gradient matching (IDGM) for domain generalization. To avoid costly second-order computations, we approximated IDGM with a simple first-order algorithm, Fish. We demonstrated our algorithm's capability to learn from invariant features (as well as ERM's failure to do so) using simple datasets such as CDSPRITES-N and the linear example. We then evaluated the model's performance on WILDS and DOMAINBED, demonstrating that Fish performs well on different subgenres of domain generalization, and surpasses baseline performance on a diverse range of vision and language tasks using different architectures such as DenseNet, ResNet-50 and BERT. Our experiments can be replicated with 1500 GPU hours on NVIDIA V100.

Despite its strong performance, similar to previous work on domain generalization, when the number of domains is large Fish struggles to outperform ERM. We are currently investigating approaches by which Fish can be made to scale to datasets with orders of magnitude more domains and expect to report on this improvement in our future work.

ETHICS STATEMENT

We believe there are no ethical concerns within this work. None of the datasets used involve human identities, and our motivation in proposing this work in no way concerns any surveillance/military use. As with many other domain generalization methods, our algorithm aims at improving the performance of machine learning systems deployed in the wilds, which can be used in many ways that will benefit the society.

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

## A  TAYLOR EXPANSION OF REPTILE AND FISH INNER-LOOP UPDATE

In this section we provide proof to Theorem 3.1. We reproduce and adapt the proof from Nichol et al. (2018) in the context of Fish, for completeness.

We demonstrate that when the inner-loop learning rate $\alpha$ is small, the direction of $G_f$ aligns with that of $G_g$, where

$$G_f = \mathbb{E}\left[(\theta - \tilde{\theta})\right] - \alpha S \cdot \bar{G}, \tag{5}$$

$$G_g = -\partial \widehat{G}/\partial \theta, \tag{6}$$

**Expanding $G_g$**  $G_g$ is the gradient of maximizing the gradient inner product (GIP).

$$G_g = -\frac{2}{S(S-1)} \sum_{\substack{i,j \in S}}^{i \neq j} \frac{\partial}{\partial \theta} G_i \cdot G_j \tag{7}$$

**Expanding $G_f$**  To write out $G_f$, we need to derive the gradient update of Fish, $\theta - \tilde{\theta}$. Let us first define some notations.

For each inner-loop with $S$ steps of gradient updates, we assume a loss functions $l$ as well as a sequence of inputs $\{d_i\}_{i=1}^S$, where $d_i := \{x_b, y_b\}_{b=1}^B$ denotes a minibatch at step $i$ randomly drawn from one of the available domains in $\{\mathcal{D}_1, \cdots, \mathcal{D}_S\}$. For reasons that will become clearer later, take extra note that the subscript $i$ here denotes the index of step, rather than the index of domain. We also define the following:

$$\widetilde{g}_i = \mathbb{E}_{d_i}\left[\frac{\partial l((x,y); \theta_i)}{\partial \theta_i}\right] \qquad \text{(gradient at step } i\text{, wrt } \theta_i\text{)} \tag{8}$$

$$\theta_{i+1} = \theta_i - \alpha \widetilde{g}_i \qquad \text{(sequence of parameters)} \tag{9}$$

$$g_i = \mathbb{E}_{d_i}\left[\frac{\partial l((x,y); \theta_1)}{\partial \theta_1}\right] \qquad \text{(gradient at step } i\text{, wrt } \theta_1\text{)} \tag{10}$$

$$H_i = \mathbb{E}_{d_i}\left[\frac{\partial^2 l((x,y); \theta_1)}{\partial \theta_1^2}\right] \qquad \text{(Hessian at initial point)} \tag{11}$$

In the following analysis we omit the expectation $\mathbb{E}_{d_i}$ and input $(x,y)$ to $l$ and instead denote the loss at step $i$ as $l_i$. Performing second-order Taylor approximation to $\widetilde{g}_i$ yields:

$$\widetilde{g}_i = l_i'(\theta_i) \tag{12}$$

$$= l_i'(\theta_1) + l_i''(\theta_1)(\theta_i - \theta_1) + \underbrace{\mathcal{O}(\|\theta_i - \theta_1\|^2)}_{=\mathcal{O}(\alpha^2)} \tag{13}$$

$$= g_i + H_i(\theta_i - \theta_1) + \mathcal{O}(\alpha^2) \tag{14}$$

$$= g_i - \alpha H_i \sum_{j=1}^{i-1} \widetilde{g}_j + \mathcal{O}(\alpha^2). \tag{15}$$

Applying first-order Taylor approximation to $\widetilde{g}_j$ gives us

$$\widetilde{g}_j = g_j + \mathcal{O}(\alpha), \tag{16}$$

plugging this back to Equation (15) yields:

$$\widetilde{g}_i = g_i - \alpha H_i \sum_{j=1}^{i-1} g_j + \mathcal{O}(\alpha^2). \tag{17}$$

For simplicity reason, let us consider performing two steps in inner-loop updates, i.e. $S = 2$. We can then write the gradient of Fish $\theta - \tilde{\theta}$ as

$$\theta - \tilde{\theta} = \alpha(\widetilde{g}_1 + \widetilde{g}_2) \tag{18}$$

$$= \alpha\underbrace{(g_1 + g_2)}_{①} - \alpha^2 \underbrace{H_2 g_1}_{②} + \mathcal{O}(\alpha^3). \tag{19}$$

Furthermore, taking the expectation of $\theta - \tilde{\theta}$ under minibatch sampling gives us (assuming independence between $g_1$ and $g_2$)

$$
\begin{aligned}
\text{①} &= \mathbb{E}_{1,2}\left[g_1 + g_2\right] = G_1 + G_2 \\
\text{②} &= \mathbb{E}_{1,2}\left[H_2 g_1\right] = \mathbb{E}_{1,2}\left[H_1 g_2\right] && \text{(interchanging indices)} \\
&= \frac{1}{2}\mathbb{E}_{1,2}\left[H_2 g_1 + H_1 g_2\right] && \text{(averaging last two eqs)} \\
&= \frac{1}{2}\mathbb{E}_{1,2}\left[\frac{\partial(g_1 \cdot g_2)}{\partial \theta_1}\right] \\
&= \frac{1}{2}\cdot\frac{\partial(G_1 \cdot G_2)}{\partial \theta_1}.
\end{aligned}
$$

Note that the only reason we can interchange the indices in ② is because the subscripts represent steps in the inner loop rather than index of domains. Plugging ①, ② in Equation (19) yields:

$$
\mathbb{E}[\theta - \tilde{\theta}] = \alpha(G_1 + G_2) - \frac{\alpha^2}{2}\cdot\frac{\partial(G_1 \cdot G_2)}{\partial \theta_1} + \mathcal{O}(\alpha^3) \tag{20}
$$

We can also expand this to the general case where $S \geq 2$:

$$
\begin{aligned}
&\mathbb{E}[\theta - \tilde{\theta}] \\
&= \alpha\sum_{s=1}^{S} G_s - \frac{\alpha^2}{S(S-1)}\sum_{i,j\in S}^{i\neq j}\frac{\partial(G_i \cdot G_j)}{\partial \theta_1} + \mathcal{O}(\alpha^3). 
\end{aligned}\tag{21}
$$

The second term in Equation (5) is $\bar{G}$, which is the full gradient of ERM defined as follow:

$$
\bar{G} = \frac{1}{S}\sum_{s=1}^{S} G_s. \tag{22}
$$

Plugging Equation (21) and Equation (22) to Equation (5) yields

$$
G_f = \mathbb{E}[\theta - \tilde{\theta}] - \alpha S \bar{G} \tag{23}
$$

$$
= -\frac{\alpha^2}{S(S-1)}\sum_{i,j\in S}^{i\neq j}\frac{\partial}{\partial \theta_1}G_i \cdot G_j \tag{24}
$$

Comparing Equation (7) to Equation (24), we have:

$$
\lim_{\alpha\to 0}\frac{G_f \cdot G_g}{\|G_f\|\,\|G_g\|} = 1.
$$

## A.1 FISH AND REPTILE: DIFFERENCES AND CONNECTIONS

As we introduced, our algorithm Fish is inspired by Reptile, a model agnostic meta-learning (MAML) algorithm.

Meta-learning aims at reducing the sample complexity of new, unseen tasks. A popular school of thinking in meta-learning is MAML, first proposed in Finn et al. (2017); Andrychowicz et al. (2016). The key idea is to backpropagate through gradient descent itself to learn representations that can be easily adapted to unseen tasks. There are close connections between meta-learning (Thrun and Pratt, 1998) and (multi-source) domain adaptation. In fact, there are a few works in domain generalization that are inspired by the meta-learning principles, such as Li et al. (2018a); Balaji et al. (2018); Li et al. (2019); Dou et al. (2019). In Ren et al. (2018), we also see the leveraging of gradient inner product in meta-learning, where it is used to determine the importance weight of training examples.

Even though meta learning and domain generalization both study $N$-way, $K$-shot problems, there are some distinct differences that set them apart. The most prominent one is that in meta learning, some examples in the test dataset will be made available at test time ($K > 0$), while in domain

**Algorithm 3** Black fonts denote steps used in **both algorithms**, colored fonts are steps unique to Fish or Reptile.

1: **for** i = 1, 2, ··· **do**
2:   $\tilde{\theta} \leftarrow \theta$
3:   Sample task $\mathcal{D}_t \sim \{\mathcal{D}_1, \cdots, \mathcal{D}_T\}$
4:   **for** $s \in \{1, \cdots, S\}$ **or** $\mathcal{D}_t \in \{\mathcal{D}_1, \cdots, \mathcal{D}_T\}$ **do**
5:     Sample batch $\boldsymbol{d}_t \sim \mathcal{D}_t$
6:     $g_t = \partial \mathcal{L}(\boldsymbol{d}_t; \tilde{\theta}) / \partial \tilde{\theta}$
7:     Update $\tilde{\theta} \leftarrow \tilde{\theta} - \alpha g_t$
8:   **end for**
9:   Update $\theta \leftarrow \theta + \epsilon(\tilde{\theta} - \theta)$
10: **end for**

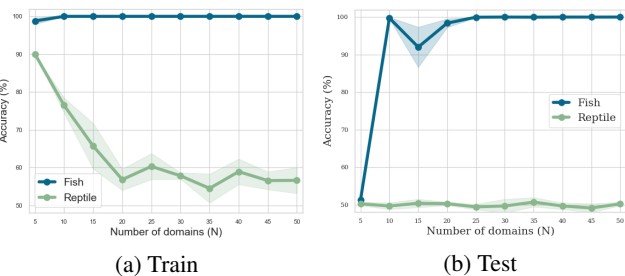

| (a) Train | (b) Test |

Figure 6: Performance on CDSPRITES-N, with $N \in [5, 50]$

generalization no example in the test dataset is seen by the model ($K = 0$); another important difference is that while domain generalization aims to train models that perform well on an unseen distribution of the *same task*, meta-learning assumes *multiple tasks* and requires the model to quickly learn an unseen task using only $K$ examples.

Due to these differences, it does not make sense in general to use MAML framework in domain generalization. As it turns out however, the idea of aligning gradients to improve generalization is relevant to both methods — The fundamental difference here that MAML algorithms such as Reptile aligns the gradients between batches from the *same* task Nichol et al. (2018), while Fish aligns those between batches from *different* tasks.

To see how this is ahiceved, let us have a look at the algorithmic comparison between Fish (blue) and Reptile (green) in Algorithm 3. As we can see, the key difference between the algorithm of Fish and Reptile is that Reptile performs its inner-loop using minibatches from the *same* task, while Fish uses minibatches from *different* tasks (*l4-8*). Based on the analysis in Nichol et al. (2018) (which we reproduce in Appendix A), this is why Reptile maximizes the within-task gradient inner products and Fish maximizes the across-task gradient inner products.

A natural question to ask here is – how does this affect their empirical performance? In Figure 6, we show the train and test performance of Fish (blue) and Reptile (green) on CDSPRITES-N. We can see that despite the algorithmic similarity between Fish and Reptile, the two methods behave very differently on this domain generalization task: while Fish's test accuracy goes to $100\%$ at $N = 10$, Reptile's test performance is always $50\%$ regardless of $N$. Moreover, we observe a dip in Reptile's training performance early on, with the accuracy plateaus at $56\%$ when $N > 20$. Reptile's poor performance on this dataset is to be expected since its inner-loop is designed to encourage within-domain generalization, which is not helpful for learning what's invariant across domains.

### A.2   WHEN $\epsilon = 1$

When we set the meta learning rate of Fish $\epsilon = 1$, Fish reduces to ERM with a data-loader that samples from only one domain in each minibatch. Note that with this change, the inner-loop step $S$ now represents the total training iterations, and as a result it increases from a constant value ($\leq 10$ in our experiments) to orders of magnitude larger (up to $10^5$ for some datasets that we use). We

demonstrate here that for this reason, the effect of maximizing inter-domain gradient inner product is significantly less prominent when $\epsilon = 1$.

To see this, let us revisit Equation (21), where we demonstrate that the expectation to Fish's update $\mathbb{E}[\theta - \tilde{\theta}]$ can be written as the sum of the following three terms via Taylor series expansion,

$$\mathbb{E}[\theta - \tilde{\theta}] = \underbrace{\alpha \sum_{s=1}^{S} G_s}_{(1)} - \underbrace{\frac{\alpha^2}{S(S-1)} \sum_{i,j \in S}^{i \neq j} \frac{\partial(G_i \cdot G_j)}{\partial \theta_1}}_{(2)} + \underbrace{\mathcal{O}(\alpha^3)}_{(3)}, \tag{25}$$

where (1) is the sum of gradients over $S$ steps of SGD updates (equivalent to ERM), (2) the gradient inner product over $S$ steps of gradients and (3) is the higher order terms of Taylor series expansion.

For both Fish and Reptile, the higher order terms are ignored under two conditions: 1) constant/non-infinite inner-loop step $S$ and 2) small learning rate $\alpha$. However, when $\epsilon = 1$, inner-loop step $S \to \infty$. As a result, the higher order terms cannot be ignored, and we can no longer conclude that gradient inner product plays an important rule in the model's updates.

## B  SMOOTHFISH: A MORE GENERAL ALGORITHM

### B.1  DERIVATION

We conclude in Appendix A that a component of Fish's update $G_f = \mathbb{E}[\theta - \tilde{\theta}] - \alpha S \cdot \bar{G}$ is in the same direction as the gradient of GIP, $G_g$. It is therefore possible to have explicit control over the scaling of the GIP component in Fish, similar to the original IDGM objective, by writing the following:

$$G_{\text{sm}} = \alpha S \cdot \bar{G} + \gamma \left( \mathbb{E}[\theta - \tilde{\theta}] - \alpha S \cdot \bar{G} \right). \tag{26}$$

By introducing the scaling term $\gamma$, we have better control on how much the objective focus on inner product vs average gradient. Note that $\gamma = 1$ recovers the original Fish gradient, and when $\gamma = 0$ the gradient $G_{\text{sm}}$ is equivalent to ERM's gradient with learning rate $\alpha S$. We name the resulting algorithm SmoothFish. See Algorithm 4.

### B.2  RESULTS

We run experiments on 4 datasets in WILDS using SmoothFish, with $\gamma$ ranging in $[0, 0.1, 0.2, 0.5, 0.8, 1, 2, 10]$. Note that when $\gamma = 0$, SmoothFish is equivalent to ERM and when $\gamma = 1$ it is equivalent to Fish. See results in Figure 7. The other hyperparameters including $\alpha$, meta steps, $\epsilon$ used here are the same as the ones used in our main experiments section.

**Algorithm 4** Smoothed version of Fish, which allows to get approximate gradients for the general form of Equation (4).

1: **for** iterations = $1, 2, \cdots$ **do**
2: $\quad \widetilde{\theta} \leftarrow \theta$
3: $\quad$ **for** $\mathcal{D}_i \in \texttt{permute}(\{\mathcal{D}_1, \mathcal{D}_2, \cdots, \mathcal{D}_S\})$ **do**
4: $\quad\quad$ Sample batch $d_i \sim \mathcal{D}_i$
5: $\quad\quad g_i = \mathbb{E}_{d_i}\left[\dfrac{\partial l((x,y); \theta)}{\partial \theta}\right]$ //Grad wrt $\theta$
6: $\quad\quad \widetilde{g}_i = \mathbb{E}_{d_i}\left[\dfrac{\partial l((x,y); \widetilde{\theta})}{\partial \widetilde{\theta}}\right]$ //Grad wrt $\widetilde{\theta}$
7: $\quad\quad$ Update $\widetilde{\theta} \leftarrow \widetilde{\theta} - \alpha \widetilde{g}_i$
8: $\quad$ **end for**
9: $\quad \bar{g} = \dfrac{1}{S}\sum_{s=1}^{S} g_i,\ g_{\text{sm}} = \alpha S \bar{g} + \gamma\left((\widetilde{\theta} - \theta) - \alpha S \bar{g}\right)$
10: $\quad$ Update $\theta \leftarrow \theta + \epsilon g_{\text{sm}}$
11: **end for**

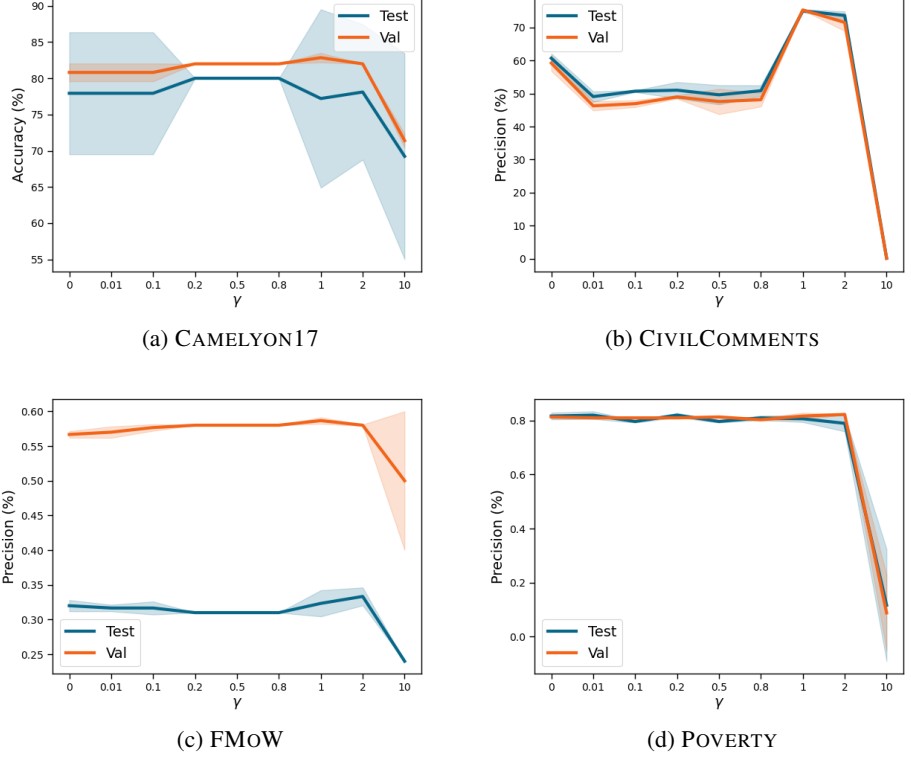

(a) CAMELYON17

(b) CIVILCOMMENTS

(c) FMoW

(d) POVERTY

Figure 7: Results on WILDS using SmoothFish with $\gamma$ ranging from 0 to 10.

## C    DISCUSSIONS AND RESULTS ON WILDS

We provide a more detailed summary of each dataset in Table 5.

Some entries in # Domains are omitted because the domains in each split overlap. Note that in this paper we report the results on WILDS v1 — the benchmark has been updated since with slightly different dataset splits. We are currently working on updating our results to v2 of WILDS.

Table 5: Details of the 6 WILDS datasets we experimented on.

| Dataset | Domains (# domains) | Data ($x$) | Target ($y$) | # Examples | | | # Domains | | |
|---|---|---|---|---|---|---|---|---|---|
| | | | | train | val | test | train | val | test |
| FMoW | Time (16), Regions (5) | Satellite images | Land use (62 classes) | 76,863 | 19,915 | 22,108 | 11, - | 3, - | 2, - |
| POVERTY | Countries (23), Urban/rural (2) | Satellite images | Asset (real valued) | 10,000 | 4,000 | 4,000 | 13, - | 5, - | 5, - |
| CAMELYON17 | Hospitals (5) | Tissue slides | Tumor (2 classes) | 302,436 | 34,904 | 85,054 | 3 | 1 | 1 |
| CIVILCOMMENTS | Demographics (8) | Online comments | Toxicity (2 classes) | 269,038 | 45,180 | 133,782 | - | - | - |
| IWILDCAM2020 | Trap locations (324) | Photos | Animal species (186 classes) | 142,202 | 20,784 | 38,943 | 245 | 32 | 47 |
| AMAZON | Reviewers (7,676) | Product reviews | Star rating (5 classes) | 1,000,124 | 100,050 | 100,050 | 5,008 | 1,334 | 1,334 |

Table 6: Results on POVERTYMAP-WILDS.

| Method | Val. Worst-U/R r | Test Worst-U/R r |
|---|---|---|
| **Fish** | 0.47 ($\pm 0.01$) | 0.30 ($\pm 0.01$) |
| IRM | 0.53 ($\pm 0.05$) | 0.43 ($\pm 0.07$) |
| ERM | 0.80 ($\pm 0.04$) | 0.78 ($\pm 0.04$) |
| ERM (ours) | 0.48 ($\pm 0.11$) | 0.29 ($\pm 0.02$) |
| Coral | 0.51 ($\pm 0.06$) | 0.45 ($\pm 0.06$) |

Table 7: Results on CAMELYON17-WILDS.

| Method | Val. Accuracy (%) | **Test Accuracy (%)** |
|---|---|---|
| **Fish** | 83.9 ($\pm 1.2$) | **74.7** ($\pm 7.1$) |
| ERM | 84.9 ($\pm 3.1$) | 70.3 ($\pm 6.4$) |
| ERM (ours) | 84.1 ($\pm 2.4$) | 70.5 ($\pm 12.1$) |
| IRM | 86.2 ($\pm 1.4$) | 64.2 ($\pm 8.1$) |
| Coral | 86.2 ($\pm 1.4$) | 59.5 ($\pm 7.7$) |

## C.1 POVERTYMAP-WILDS

*Task: Asset index prediction (real-valued). Domains: 23 countries*

The task is to predict the real-valued asset wealth index of an area, given its satellite imagery. Since the number of domains considered here is large (23 countries), instead of looping over all $S$ domains in each inner-loop, we sample $N << S$ domains in each iteration and perform inner-loop updates using minibatches from these domains only to speed up computation. For this dataset we choose $N = 5$ by hyper-parameter search.

**Evalutaion:** *Worst-U/R Pearson Correlation (r).* Following the practice in WILDS benchmark, we compare the results by computing the worst region Pearson correlation (r) between the predicted and ground-truth asset index over 3 random seed runs.

**Results:** We train the model using a ResNet-18 (He et al., 2016) backbone. See Table 6.

We see that Fish obtains the highest test performance, with the same validation performance as the best baseline. The performance is more stable between validation and test, and the standard deviation is smaller than for the baselines. We also report the results of ERM models trained in our environment as "ERM (ours)", which shows similar performance to the canonical results reported in the WILDS benchmark itself ("ERM").

## C.2 CAMELYON17-WILDS

*Task: Tumor detection (2 classes). Domains: 5 hospitals*

The CAMELYON17-WILDS dataset contains 450,000 lymph-node scans from 5 hospitals. Due to the size of the dataset, instead of training with Fish from scratch, we pre-train the model with ERM using the recommended hyper-parameters in Koh et al. (2020), and fine-tune with Fish. For this dataset, we find that Fish performs the best when starting from a pretrained model that has not yet converged, achieving much higher accuracy than the ERM model. we provide an ablation study on this in Appendix E.

**Evaluation:** *Average accuracy.* We evaluate the average accuracy of this binary classification task. Following Koh et al. (2020), we show the mean and standard deviation of results over 10 random seeds runs. The number of random seeds required here is greater than other WILDS datasets due to the large variance observed in results. Note that these random seeds are not only applied during the fine-tuning stage, but also to the pretrained models to ensure a fair comparison.

**Results:** Following the practice in WILDS, we adopt DenseNet-121's (Huang et al., 2017) architecture for models trained on this dataset. See results in Table 7.

The results show that Fish significantly outperforms all baselines – its test accuracy surpasses the best performing baseline by 6%. Also note that for all other baselines, there is a large gap between validation and test accuracy ($11\% \sim 27\%$). This is because WILDS chose the hospital that is the most

difficult to generalize to as the test split to make the task more challenging. Surprisingly, as we can observe in Table 7, the discrepancy between test and validation accuracy of Fish is quite small (3%). The fact that it is able to achieve a similar level of accuracy on the worst-performing domain further demonstrates that Fish does not rely on domain-specific information, and instead makes predictions using the invariant features across domains.

To demonstrate that Fish's strong performance on CAMELYON17's selected test set is not merely coincidental, we randomly chose 3 ways of assigning the 5 domains as train/test/val (3/1/1 domains) splits. See results collected over 10 random seeds in Table 8.

Notably, on these different splits of CAMELYON17 Fish still on average outperforms ERM (with only 2 exceptions), demonstrating that Fish can generally achieve good performance on the Camelyon17 dataset.

Table 8: Results on CAMELYON17-WILDS with shuffled splits.

| Train/Val/Test ID | Val, ERM (%) | Val, Fish (%) | Test, ERM (%) | Test, Fish (%) |
|---|---|---|---|---|
| 012/3/4 | 71.3 ($\pm6.3$) | 71.9 ($\pm5.1$) | 80.9 ($\pm8.6$) | 88.9 ($\pm6.9$) |
| 124/0/3 | 72.3 ($\pm5.3$) | 74.1 ($\pm4.3$) | 63.4 ($\pm6.6$) | 65.9 ($\pm3.9$) |
| 234/0/1 | 83.9 ($\pm3.8$) | 84.1 ($\pm2.3$) | 75.6 ($\pm2.8$) | 73.9 ($\pm3.5$) |
| 034/1/2 (original) | 84.3 ($\pm2.1$) | 82.5 ($\pm1.2$) | 73.3 ($\pm9.0$) | 79.5 ($\pm6.0$) |

Table 9: Results on FMoW-WILDS.

| Method | Val. Accuracy (%) | | Test Accuracy (%) | |
|---|---|---|---|---|
| | Average | Worst | Average | Worst |
| **Fish** | 57.8 ($\pm0.15$) | 49.5 ($\pm2.34$) | 51.8 ($\pm0.32$) | **34.6 ($\pm0.18$)** |
| ERM | 59.5 ($\pm0.37$) | 48.9 ($\pm0.62$) | 53.0 ($\pm0.55$) | 32.3 ($\pm1.25$) |
| ERM (ours) | 59.9 ($\pm0.22$) | 47.1 ($\pm1.21$) | 52.9 ($\pm0.18$) | 30.9 ($\pm1.53$) |
| IRM | 57.4 ($\pm0.37$) | 47.5 ($\pm1.57$) | 50.8 ($\pm0.13$) | 30.0 ($\pm1.37$) |
| Coral | 56.9 ($\pm0.25$) | 47.1 ($\pm0.43$) | 50.5 ($\pm0.36$) | 31.7 ($\pm1.24$) |

Table 10: Results on CIVILCOMMENTS-WILDS.

| Method | Val. Accuracy (%) | | Test Accuracy (%) | |
|---|---|---|---|---|
| | Average | Worst | Average | Worst |
| **Fish** | 88.8 ($\pm0.6$) | 70.5 ($\pm1.0$) | 89.4 ($\pm0.2$) | **75.3 ($\pm0.6$)** |
| Group DRO | 89.6 ($\pm0.3$) | 68.7 ($\pm1.0$) | 89.4 ($\pm0.3$) | 70.4 ($\pm2.1$) |
| Reweighted | 89.1 ($\pm0.3$) | 67.9 ($\pm1.2$) | 88.9 ($\pm0.3$) | 67.3 ($\pm0.1$) |
| ERM | 92.3 ($\pm0.6$) | 53.6 ($\pm0.7$) | 92.2 ($\pm0.6$) | 58.0 ($\pm1.2$) |
| ERM (ours) | 92.1 ($\pm0.5$) | 54.1 ($\pm0.4$) | 92.5 ($\pm0.3$) | 58.1 ($\pm1.7$) |

## C.3 FMoW-WILDS

*Task: Infrastructure classification (62 classes). Domains: 80 (16 years x 5 regions)*

Similar to CAMELYON17-WILDS, since the number of domains is large, we sample $N = 5$ domains for each inner-loop. To speed up computation, we also use a pretrained ERM model and fine-tune with Fish; different from Appendix C.2, we find the best-performing models are acquired when using converged pretrained models (see details in Appendix E).

**Evaluation:** *Average & worst-region accuracies.* Following WILDS, the average accuracy evaluates the model's ability to generalize over years, and the worst-region accuracy measures the model's performance across regions under a time shift. We report results using 3 random seeds.

**Results:** Following Koh et al. (2020), we use a DenseNet-121 pretrained on ImageNet for this dataset. Results in Table 9 show that Fish has the highest worst-region accuracy on both test and validation sets. It ranks second in terms of average accuracy, right after ERM. Again, Fish's performance is notably stable with the smallest standard deviation across all metrics compared to baselines.

## C.4 CIVILCOMMENTS-WILDS

*Task: Toxicity detection in online comments (2 classes). Domains: 8 demographic identities.*

The CIVILCOMMENTS-WILDS contains 450,000 comments collected from online articles, each annotated for toxicity and the mentioning of demographic identities. Again, we use ERM pre-trained model to speed up computation, and sample $N = 5$ domains for each inner-loop.

**Evaluation:** *Worst-group accuracy.* To study the bias of annotating comments that mentions demographic groups as toxic, the WILDS benchmark proposes to evaluate the model's performance

by doing the following: 1) Further separate each of the 8 demographic identities into 2 groups by toxicity – for example, separate *black* into *black, toxic* and *black, not toxic*; 2) measure the accuracies of these $8 \times 2 = 16$ groups and use the lowest accuracy as the final evaluation of the model. This metric is equivalent to computing the sensitivity and specificity of the classifier on each demographic identity, and reporting the worse of the two metrics over all domains. Good performance on the group with the worst accuracy implies that the model does not tend to use demographic identity as an indicator of toxic comments.

Again, following Koh et al. (2020) we report results of 3 random seed runs.

**Results:** We compare results to the baselines used in the WILDS benchmark over 3 random seed runs in Table 10. All models are trained using DistilBERT (Sanh et al., 2019).

The results show that Fish outperforms the best baseline by $4\%$ and $7\%$ on the test and validation set's worst-group accuracy respectively, and is competitive in terms of average accuracy with ERM (within standard deviation). The strong performance of Fish on worst-group accuracy suggests that the model relies the least on demographic identity as an indicator of toxic comments compared to other baselines. ERM, on the other hand, has the highest average accuracy and the lowest worst-group accuracy. This indicates that it achieves good average performance by leveraging the spurious correlation between toxic comments and the mention of certain demographic groups.

Note that different from all other datasets in WILDS that focus on *pure domain generalization* (i.e, no overlap between domains in train and test splits), CIVILCOMMENTS-WILDS is a *subpopulation shift* problem, where the domains in test are a subpopulation of the domains in train. As a result, the baseline models used in WILDS for this dataset are different from the methods used in all other datasets, and are tailored to avoiding systematic failure on data from minority subpopulations. Fish works well in this setting too without any changes or special sampling strategies (such as $*$ and $+$ in Table 10). This further demonstrates the good performance of our algorithm on different domain generalization scenarios.

### C.5 IWILDCAM-WILDS

*Task: Animal species (186 classes). Domains: 324 camera locations.*

The dataset consists of over 200,000 photos of animal in the wild, using stationary cameras across 324 locations. Classifying animal species from these heat or motion-activated photos is especially challenging: methods can easily rely on the background information of photos from the same camera setup. Fish models are pretrained with ERM till convergence, and for each inner loop we sample from $N = 10$ domains.

**Evaluation:** *Macro F1 score*. Across the 186 class labels, we report average accuracy and both weighted and macro F1 scores (F1-w and F1-m, respectively, in Table 11). We run 3 random seeds for each model.

**Results:** All models reported in Table 7 are trained using a ResNet-50. We find Fish to outperform baselines on both test accuracy and weighted F1, with a $1\%$ improvement on both metrics over the best performing model (ERM). However, this comes at the cost of lower macro F1 score, where Fish performs $1\%$ worse than ERM models that we trained and $3\%$ than the ERM reported in WILDS. This suggests that Fish is less good at classifying rarer species, however the overall accuracy on the test dataset is improved.

Although Fish did not outperform the ERM baseline on the primary evaluation metric proposed in Koh et al. (2020), we found the improvement of Fish in both accuracy and weighted F1 to be robust across a range of hyperparameters. See more details on this in Appendix D.

### C.6 AMAZON-WILDS

*Task: Sentiment analysis (5 classes). Domains: 7,676 Amazon reviewers.*

The dataset contains 1.4 million customer reviews on Amazon from 7,676 customers, and the task is to predict the score (1-5 stars) given the review. Similarly, we pretrained the model with ERM

till convergence, and due to the large number of domains ($S = 5008$ in train) we sample $N = 5$ reviewers for each inner loop.

**Evaluation:** *10th percentile accuracy*. Reporting the accuracy of the 10th percentile reviewer helps us assess whether the model performance is consistent across different reviewers. The results in Table 12 are reported over 3 random seeds.

**Results:** The model is trained using DISTILBERT (Sanh et al., 2019) backbone. While Fish has lower average accuracy compared to ERM, its 10th percentile accuracy matches that of ERM, outperforming all other baselines.

Table 11: Results on IWILDCAM-WILDS.

| Method | Test ID Macro F1 (%) | Test ID Avg Acc (%) | Test OOD Macro F1 (%) | Test OOD Avg Acc (%) |
|---|---|---|---|---|
| Fish | 40.3 (±0.6) | 73.8 (±0.1) | 22.0 (±1.8) | 64.7 (±2.6) |
| GroupDRO | 37.5 (±1.7) | 71.6 (±2.7) | 23.9 (±2.1) | 72.7 (±2.0) |
| ERM | 47.0 (±1.4) | 75.7 (±0.3) | 31.0 (±1.3) | 71.6 (±2.5) |
| Coral | 43.5 (±3.5) | 73.7 (±0.4) | 32.8 (±0.1) | 73.3 (±4.3) |
| IRM | 22.4 (±7.7) | 59.9 (±8.1) | 15.1 (±4.9) | 59.8 (±3.7) |

Table 12: Results on AMAZON-WILDS.

| Method | Val. Accuracy (%) | | Test Accuracy (%) | |
|---|---|---|---|---|
| | Average | 10-th per. | Average | **10-th per.** |
| **Fish** | 72.5 (±0.0) | 54.0 (±0.0) | 71.7 (±0.0) | **53.3 (±0.0)** |
| **ERM** | 72.7 (±0.1) | 55.2 (±0.7) | 71.9 (±0.1) | **53.8 (±0.0)** |
| IRM | 71.5 (±0.3) | 54.2 (±0.8) | 70.5 (±0.3) | 52.4 (±0.8) |
| Reweighted | 69.1 (±0.0) | 52.1 (±0.2) | 68.6 (±0.6) | 52.0 (±0.0) |

## D  HYPERPARAMETERS

In Table 13 we list the hyperparameters we used to train ERM. The same hyperparameters were used for producing ERM baseline results and as pretrained models for Fish. In `val. metric` we report the metric on validation set that is used for model selection, and in `cut-off` we specify when to stop training when using ERM to generate pretrained models.

Table 13: Hyperparameters for ERM. We follow the hyperparameters used in WILDS benchmark. Note that we did not use a pretrained model for POVERTY, therefore its cut-off condition is not reported.

| Dataset | Model | Learning rate | Batch size | Weight decay | Optimizer | Val. metric | Cut-off |
|---|---|---|---|---|---|---|---|
| CAMELYON17 | Densenet-121 | 1e-3 | 32 | 0 | SGD | acc. avg. | iter 500 |
| CIVILCOMMENTS | DistilBERT | 1e-5 | 16 | 0.01 | Adam | acc. wg. | Best val. metric |
| FMoW | Densenet-121 | 1e-4 | 64 | 0 | Adam | acc. avg. | Best val. metric |
| IWILDCAM | Resnet-50 | 1e-4 | 16 | 0 | Adam | F1-macro (all) | Best val. metric |
| POVERTY | Resnet-18 | 1e-3 | 64 | 0 | Adam | Worst-U/R Pearson (r) | - |
| AMAZON | DistilBERT | 2e-6 | 8 | 0.01 | Adam | 10th percentile acc. | - |

In Table 14 we list out the hyperparameters we used to train Fish. Note that we train Fish using the same model, batch size, val metric and optimizer as ERM – these are not listed in Table 14 to avoid repetitions. Weight decay is always set as 0.

## E  ABLATION STUDIES ON PRE-TRAINED MODELS

In this section we perform ablation study on the convergence of pretrained ERM models. Note that the pretraining is done on the same domain generalization dataset that Fish is trained on later, not on ImageNet. We study the performance of Fish with the following three configurations of pretrained ERM models:

Table 14: Hyperparameters for Fish.

| Dataset | Group by | $\alpha$ | $\epsilon$ | # domains | Meta steps |
|---|---|---|---|---|---|
| CAMELYON17 | Hospitals | 1e-3 | 0.01 | 3 | 3 |
| CIVILCOMMENTS | Demographics × toxicity | 1e-5 | 0.05 | 16 | 5 |
| FMoW | time × regions | 1e-4 | 0.01 | 80 | 5 |
| IWILDCAM | Trap locations | 1e-4 | 0.01 | 324 | 10 |
| POVERTY | Countries | 1e-3 | 0.1 | 23 | 5 |
| AMAZON | Reviewers | 2e-6 | 0.01 | 7,676 | 5 |

1. Model is trained on $10\%$ of the data (epoch 1);

2. Model is trained on $50\%$ of the data (epoch 1);

3. Model at convergence.

By comparing the results between these three settings, we demonstrate how the level of convergence affects the Fish's training performance. See results in Table 15. Note that POVERTY is excluded here because the dataset is small enough that we are able to train Fish from scratch.

Table 15: Ablation study on pretrained ERM models.

| Model | FMoW | CAMELYON17 | IWILDCAM | CIVILCOMMENTS |
|---|---|---|---|---|
| | Test Avg Acc | Test Avg Acc | Test Macro F1 | Test Worst Acc |
| 10% data | 21.7 ($\pm 2.5$) | 79.1 ($\pm 12.3$) | 13.7 ($\pm 0.5$) | 71.8 ($\pm 1.3$) |
| 50% data | 31.0 ($\pm 0.8$) | 64.6 ($\pm 12.3$) | 19.0 ($\pm 0.06$) | 74.2 ($\pm 0.5$) |
| Converged | 32.7 ($\pm 1.2$) | 63.5 ($\pm 8.2$) | 23.7 ($\pm 0.9$) | 73.8 ($\pm 1.8$) |

We see that CIVILCOMMENTS sustain good performance using pretrained models at different convergence levels. FMoW and IWILDCAM on the other hand seems to have strong preference towards converged model, and the results worsen as the amount of data seen during training goes down. CAMELYON17 achieves the best performance when only $10\%$ of data is seen, and the test accuracy decreases while training with models with higher level of convergence.

## F  ABLATION STUDIES ON HYPERPARAMETERS

$\alpha$ **and** $\epsilon$   We study the effect of changing Fish's inner loop learning rate $\alpha$ and outer loop learning rate $\epsilon$. To make the comparisons more meaningful, we keep $\alpha \cdot \epsilon$ constant while changing their respective values. See results in Figure 8.

**Meta steps** $N$   For most of the datasets we studied (all apart from CAMELYON17 where $T = 3$) we sample a $N$-sized subset of all $T$ domains available for training (see Table 14 for $T$ of each dataset). Here we study when $N = 5, 10, 20$.

Table 16: Ablation study on meta steps $N$.

| $N$ | FMoW | POVERTY | IWILDCAM | CIVILCOMMENTS |
|---|---|---|---|---|
| | Test Avg Acc | Test Pearson r | Test Macro F1 | Test Worst Acc |
| 5 | 33.0 ($\pm 1.6$) | 80.3 ($\pm 1.7$) | 23.7 ($\pm 0.9$) | 74.3 ($\pm 1.5$) |
| 10 | 32.7 ($\pm 1.2$) | 80.0 ($\pm 0.8$) | 23.7 ($\pm 0.5$) | 73.4 ($\pm 1.0$) |
| 20 | 33.3 ($\pm 2.1$) | 77.7 ($\pm 2.1$) | 23.7 ($\pm 0.9$) | 72.6 ($\pm 2.3$) |

In general altering these hyperparameters don't have a huge impact on the model's performance, however it does seem thet when $N = 20$ the performance on some datasets (POVERTY and CIVIL-COMMENTS) degrade slightly.

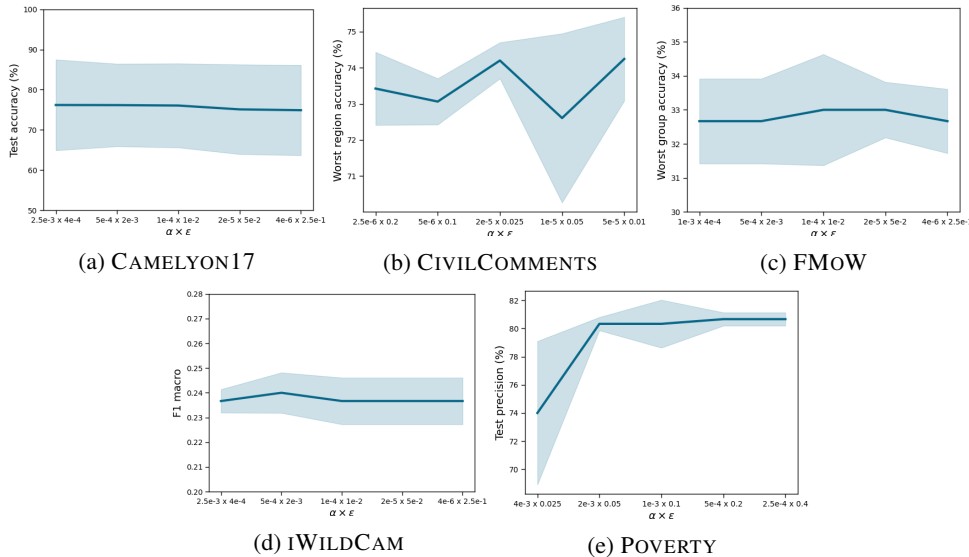

Figure 8: Ablation studies on $\alpha$ and $\epsilon$. Note that $\alpha \times \epsilon$ remains constant in all experiments, and the midpoint of each plot is the hyperparameter we chose to use to report our experiment results.

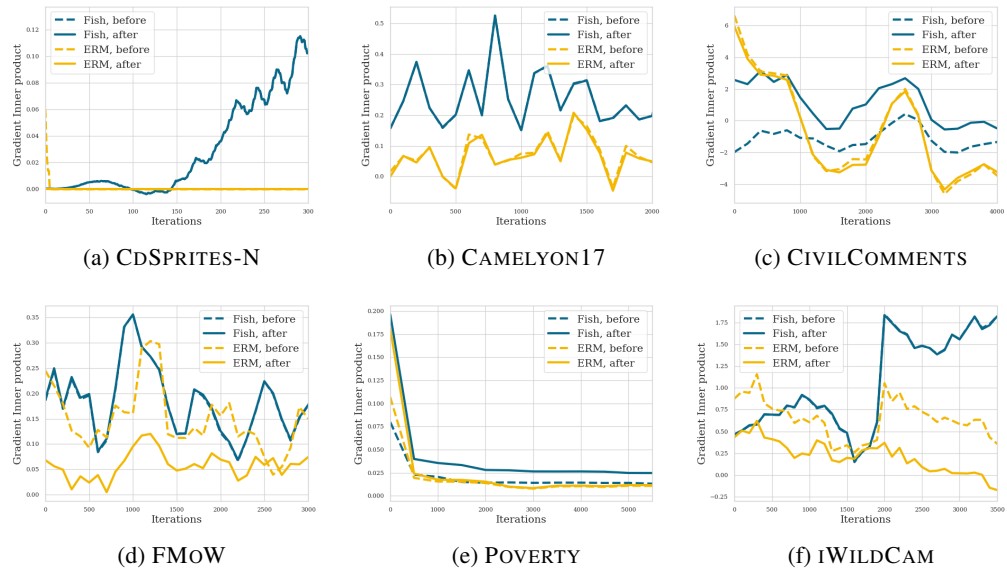

Figure 9: Gradient inner product values during the training for CDSPRITES-N (N=15) and 5 different WILDS datasets.

## G  TRACKING GRADIENT INNER PRODUCT

In Figure 9, we demonstrate the progression of inter-domain gradient inner products during training using different objectives. We train both **Fish** (blue) and **ERM** (yellow) untill convergence while recording the normalized gradient inner products (i.e. cosine similarity) between minibatches from different domains used in each inner-loop. The gradient inner products are computed both before (dotted) and after (solid) the model's update. To ensure a fair comparison, we use the exact same sequence of data for Fish and ERM (see Appendix I for more details).

Inevitably, the gradient inner product trends differently for each dataset since the data, types of domain splits and the choice of architecture are very different. In fact, the plot for CDSPRITES-N and POVERTY are significantly different from others, with a dip in gradient inner product at the beginning

of training – this is because these are the two datasets that we train from scratch. On all other datasets, the gradient inner products are recorded when fine-tuning with Fish.

Despite their differences, there are some important commonalities between these plots: if we compare the pre-update (dotted) to post-update (solid) curves, we can see that ERM updates often result in the decrease of gradient inner product, while for Fish it can either increase significantly (Figure 9c and Figure 9e) or at least stay at the same level (Figure 9a, Figure 9b, Figure 9d and Figure 9f). As a result of this, we can see that the post-update gradient inner product of Fish is always greater than ERM at convergence.

The observations here shows that Fish is effective in increasing/maintaining the level of inter-domain gradient inner product and sheds some lights on its efficiency on the datasets we studied.

## H    SUMMARY OF DOMAINBED

DOMAINBED is a testbed for domain generalization that implements consistent experimental protocols across SOTA methods to ensure fair comparison. It contains 7 popular domain generalization datasets, including Colored MNIST (Arjovsky et al., 2019), Rotated MNIST (Ghifary et al., 2015), VLCS (Fang et al., 2013), PACS (Li et al., 2017), OfficeHome (Venkateswara et al., 2017), Terra Incognita (Beery et al., 2018) and DomainNet (Peng et al., 2019), and offers comparison to a variety of SOTA domain generalization methods, including IRM (Arjovsky et al., 2019), Group DRO (Hu et al., 2018; Sagawa et al., 2019), Mixup (Yan et al., 2020), MLDG (Li et al., 2018a), Coral (Sun and Saenko, 2016), MMD (Li et al., 2018b), DANN (Ganin et al., 2016) and CDANN (Li et al., 2018).

## I    ALGORITHM FOR TRACKING GRADIENT INNER PRODUCT

To make sure that the gradients we record for ERM and Fish are comparable, we use the same sequence of $S$-minibatches to train both algorithms. See Algorithm 6 for details.

---

**Algorithm 5** Function **GIP**

1: **function GIP**$(\{d_1, d_2, \cdots, d_N\}, \theta)$:
2: **for** $d_n \in \{d_1, d_2, \cdots, d_N\}$ **do**
3:     $g_n = \partial l(\boldsymbol{d}_n; \theta)/\partial \theta$
4: **end for**
5: $\bar{\bar{\boldsymbol{g}}} = \frac{1}{S(S-1)} \sum_{i,j \in S}^{i \neq j} g_i \cdot g_j$
6: **return** $\bar{\bar{\boldsymbol{g}}}$

---

## J    T-SNE VISUALISATION OF LEARNED REPRESENTATIONS

In this section we visualize the representation learned by Fish and ERM with t-SNE (van der Maaten and Hinton, 2008) for PACS, VLCS and OfficeHome datasets. See Figure 10.

We can see that Fish and ERM are both capable of forming distinctive label-clusters on PACS (first row) and VLCS (second row), however with ERM we can observe within each label cluster that sub-clusters of domains are forming. This is particularly the case for the the red and yellow cluster in Figure 10a and cyan cluster in Figure 10c. The domain clustering phenomena is not observed for Fish. On the other hand, for the OfficeHome dataset where Fish outperforms ERM by more than $1\%$, we clearly see that Fish exhibits better label clustering performance than ERM.

---

**Algorithm 6** Algorithm of collecting gradient inner product $\bar{\bar{g}}$ for Fish and ERM both before and after updates. See **GIP** in Algorithm 5.

---

1: Initialize Fish $\theta_f \leftarrow \theta$, ERM $\theta_e \leftarrow \theta$
2: **for** i = $1, 2, \cdots$ **do**
3:     // Get all minibatches
4:     **for** $\mathcal{D}_n \in \{\mathcal{D}_1, \mathcal{D}_2, \cdots, \mathcal{D}_N\}$ **do**
5:        Sample batch $\boldsymbol{d}_n \sim \mathcal{D}_n$
6:     **end for**
7:     // GradInnerProd before update
8:     $\bar{\bar{\boldsymbol{g}}}_{Fb} = \textbf{GIP}(\{d_1, d_2, \cdots, d_N\}, \theta_f)$
9:     $\bar{\bar{\boldsymbol{g}}}_{Eb} = \textbf{GIP}(\{d_1, d_2, \cdots, d_N\}, \theta_e)$
10:    // Fish training
11:    $\tilde{\theta} \leftarrow \theta_f$
12:    **for** $d_n \in \{d_1, d_2, \cdots, d_N\}$ **do**
13:       $g_n = \partial l(\boldsymbol{d}_n; \tilde{\theta})/\partial\tilde{\theta}$
14:       Update $\tilde{\theta} \leftarrow \tilde{\theta} - \alpha g_n$
15:    **end for**
16:    $\theta_f \leftarrow \theta_f + \epsilon(\tilde{\theta} - \theta_f)$
17:    // Rearrange minibatches
18:    $d = \texttt{shuffle}(\texttt{concat}(d_1, d_2, \cdots, d_N))$
19:    $\{\tilde{d}_1, \tilde{d}_2, \cdots, \tilde{d}_N\} = \texttt{split}(d)$
20:    // ERM training
21:    **for** $\tilde{d}_n \in \{\tilde{d}_1, \tilde{d}_2, \cdots, \tilde{d}_N\}$ **do**
22:       $g_n = \partial l(\tilde{d}_n; \theta_e)/\partial\theta_e$
23:       Update $\theta_e \leftarrow \theta_e - \alpha g_n$
24:    **end for**
25:    // GradInnerProd after update
26:    $\bar{\bar{\boldsymbol{g}}}_{Fa} = \textbf{GIP}(\{d_1, d_2, \cdots, d_N\}, \theta_f)$
27:    $\bar{\bar{\boldsymbol{g}}}_{Ea} = \textbf{GIP}(\{d_1, d_2, \cdots, d_N\}, \theta_e)$
28: **end for**
29: **Return** $\bar{\bar{\boldsymbol{g}}}_{Fb}, \bar{\bar{\boldsymbol{g}}}_{Fa}, \bar{\bar{\boldsymbol{g}}}_{Eb}, \bar{\bar{\boldsymbol{g}}}_{Ea}$

---

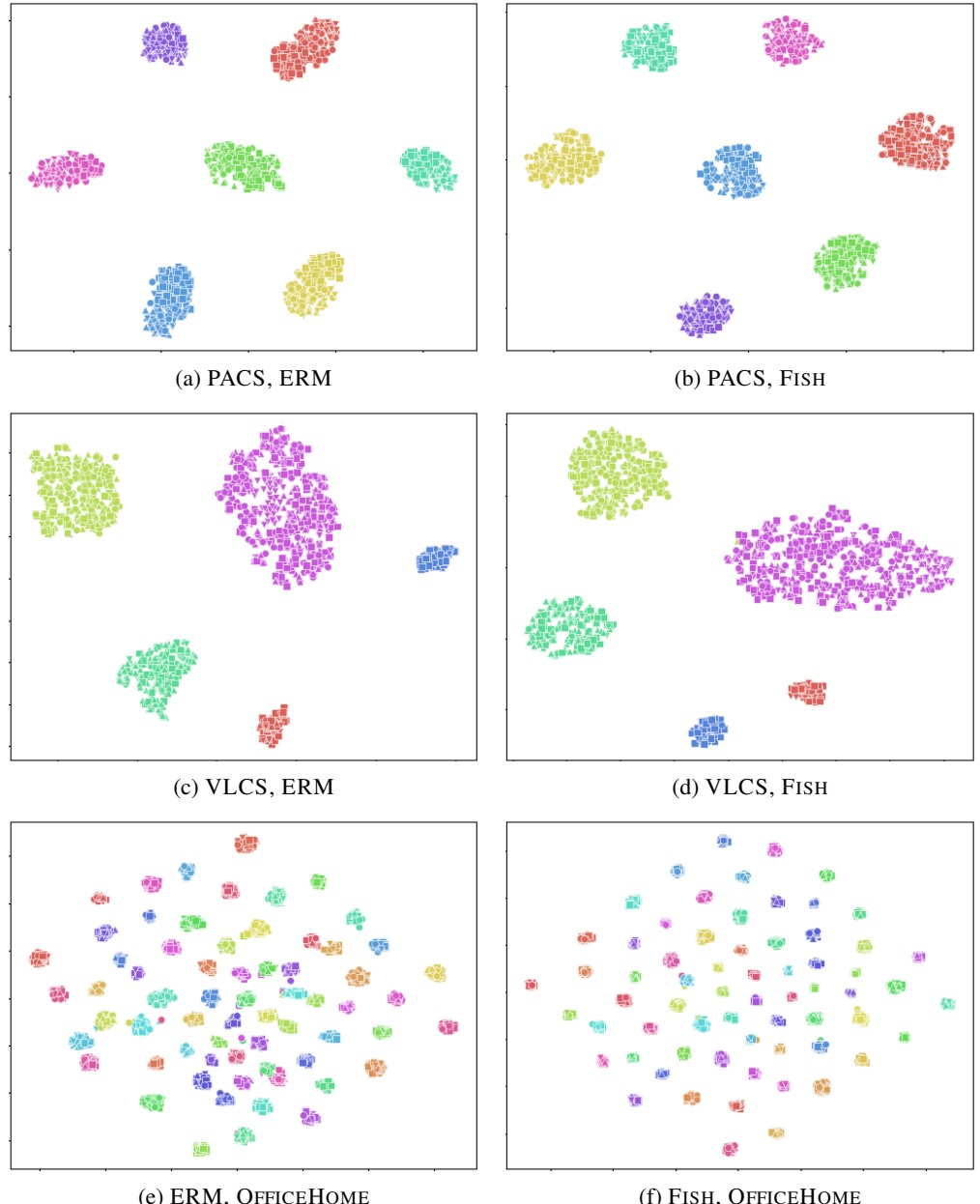

Figure 10: t-SNE plot for PACS, VLCS and OfficeHome. Colors represent labels, markers (shape of each datapoint) represent domain.

