# OpenReview forum: "Gradient Matching for Domain Generalization"
_ICLR.cc/2022/Conference — ICLR 2022 Poster_

### Official Review · Reviewer_eig7 · 2021-11-02

**Correctness:** 3
**Technical Novelty And Significance:** 3
**Empirical Novelty And Significance:** 3
**Recommendation:** 6
**Confidence:** 4

**Main Review:**

Pros:

(1) The theorem of FISH looks interesting, which can avoid costly second-order computations.

(2) The pitfall of ERM is interesting to explain why ERM fail in the domain generalization.

(3) This paper is well written and easy to follow.

(4) Experiments on the WILDs and DomainBed demonstrate the advantages of the proposed methods in a practical.

Cons:

(1) The motivation for using gradient matching is not clear.  e.g., Why using gradient matching can achieve domain-invariant representations?

(2) Lack of comparative experiments of direct optimization of IDGM on the DomainBed.

(3) Some sentences are not accurately described. For example: In the related work 5. Gradient alignment: "While they achieve the best performance on the training set, both IGA and ERM could completely fail when generalizing to unseen domains."  Not all ERM algorithms fail [1][2].

(4)The structure of the inner-loop and outer-loop is similar to MAML or Reptile. Need more explanation and the difference of MAML.



Reference:

[1] Gulrajani et al 2020. In search of lost domain generalization.

[2] Cha, Junbum, et al. "Swad: Domain generalization by seeking flat minima." arXiv preprint arXiv:2102.08604 4 (2021).








**Summary Of The Paper:**

This paper proposed inter-domain gradient matching for domain generalization. They also approximated the proposed model with a simple first-order algorithm to avoid costly second-order computations.  The performance on the WILDs and DomainBed seems better than the ERM algorithm.

**Summary Of The Review:**

As stated in the main reviews, the motivation of using a matching gradient is not particularly impressive, but overall the paper does provide some interesting analysis that may encourage new ways of thinking about domain generalization problems. As such, I think this paper may be of interest to the ICLR community.

---

> ### Author Response · Authors · 2021-11-15
> **Response to reviewer**
>
> > The motivation for using gradient matching is not clear. e.g., Why using gradient matching can achieve domain-invariant representations?
>
> Our motivation is the following: when the gradient directions of two domains align, improving the performance on one domain via gradient descent would also improve the performance on the other.
>
>
> We discuss the motivation of our objective in details in section 3.3 and provide empirical analysis in the last paragraph of section 4.4. We also point out in both related work and section 4.4 that concurrent to our work are [1] and [2] which also uses gradient alignment to achieve invariant predictions. We'd be happy to answer any further question the reviewer has about this.
>
>
> > Lack of comparative experiments of direct optimization of IDGM on the DomainBed.
>
> As we mentioned in section 4.4, training with IDGM on non-toy datasets is too expensive. As an example, a model with IDGM objective trained on VLCS dataset with batch size 32 and standard ResNet18 backbone does not fit a 32GB GPU (NVIDIA V100). We did however, evaluate IDGM objective on toy datasets such as the linear example (section 3) and the color-dsprites dataset (section 4.1).
>
>
> > Some sentences are not accurately described. For example: In the related work 5. Gradient alignment: "While they achieve the best performance on the training set, both IGA and ERM could completely fail when generalizing to unseen domains." Not all ERM algorithms fail.
>
> Thank you for your suggestion! We did not mean to indicate that ERM always fail, only that it *could* fail in some cases, as highlighted with our linear example in Section 2. We will clarify this in our updated manuscript.
>
>
> > The structure of the inner-loop and outer-loop is similar to MAML or Reptile. Need more explanation and the difference of MAML.
>
> Our work's inner-loop construction does share some similarity to Reptile, however Reptile significantly underperform on domain generalisation tasks -- for more details please refer to Appendix A.1, where we discuss both the conceptual and empirical difference between Fish and Reptile.
>
> **References:**
>
> [1] Parascandolo, A. Neitz, A. Orvieto, L. Gresele, and B. Schölkopf. Learning explanations that are hard to vary. In ICLR 2021: The Ninth International Conference on Learning Representations, 2021.
>
> [2] M. Koyama and S. Yamaguchi. Out-of-distribution generalization with maximal invariant predictor. In arXiv e-prints, 2021.

---

> > ### Comment · Reviewer_eig7 · 2021-11-25
> > **Reply**
> >
> > Many thanks to the authors for the response and clarification, this is very helpful.
> >
> >  For the motivation, thanks for the clarification, this is indeed much better.
> >
> >  In sum, most of my concerns are addressed. I will keep my score and tend to accept this paper.

---

### Official Review · Reviewer_t4Z6 · 2021-11-03

**Correctness:** 4
**Technical Novelty And Significance:** 3
**Empirical Novelty And Significance:** 2
**Recommendation:** 6
**Confidence:** 5

**Main Review:**

Strength:

1) Authors motivate a proposed method clearly with their illustration in figure 2 and explanation in section 3.3

2) Contribution is a novel combination of existing ideas. Authors worked out the Taylor series expansion of the gradient and came up with a linear approximation. Authors modified the Reptile algorithm which works on improving the GIP inside a particular task (or domain) to improve the GIP across different tasks (or domains)

3) By the virtue of their simplification, their model update process is much cleaner than it would have been had they used IDGM itself

4) This linear approximation also helps to not worry about saving intermediate gradients of all the past steps as might be the case in Reptile.

Weakness:

1) It is not clear if the improvements given by the proposed method are statistically significant. It is also not clear why this method performs better in certain datasets and worse on others.

2) Missing literature.

3) Comparing Fish and IDGM might be necessary to confirm that the approximation works.

Review:

1) How is the concept of using gradient inner product different from using kernel methods? Can authors compare and contrast with a popular DG method [1]?

2) Is there any way one can check or compare - how well the linear approximation is working? If given infinite resources, can authors improve their scores further? If authors can show this on at least a few dataset, it will really help to understand the method better.

3) iWildCam is a much tougher dataset to work with where there are pictures of deers in grass (for ex) which is present throughout all the images. Similar vegetation is also present around other wild animals in other pictures. This aspect might have confused this approach significantly as it might think vegetation to be an invariant feature. This is where proposed method doesn’t perform too well. Can authors comment why?

4) Gradient inner products are also related to Neural Tangent Kernels [2]. Can authors comment more on this?

[1] Blanchard, Gilles, Aniket Anand Deshmukh, Ürün Dogan, Gyemin Lee, and Clayton Scott. "Domain Generalization by Marginal Transfer Learning." arXiv preprint arXiv:1711.07910 (2017) J. Mach. Learn. Res. (JMLR) 22 (2021): 2-1.

[2] Jacot, Arthur, Franck Gabriel, and Clément Hongler. "Neural tangent kernel: Convergence and generalization in neural networks." arXiv preprint arXiv:1806.07572 (2018).



**Summary Of The Paper:**

Authors propose an inter-domain gradient matching objective that targets domain generalization by maximizing the inner product between gradients from different domains. Authors also give a computationally efficient optimization for the proposed method and give results on various datasets.

**Summary Of The Review:**

Literature could be improved. For better understanding slow but exact algorithm could be tested on small dataset.

---

> ### Author Response · Authors · 2021-11-15
> **Response to reviewer**
>
>
>
> > iWildCam is a much tougher dataset to work with where there are pictures of deers in grass (for ex) which is present throughout all the images. Similar vegetation is also present around other wild animals in other pictures. This aspect might have confused this approach significantly as it might think vegetation to be an invariant feature. This is where proposed method doesn’t perform too well. Can authors comment why?
>
> It is indeed true that iWildCam is a complicated dataset to work with. Apart from the reason aptly pointed out by the reviewer, we believe that it is also challenging due to the large number of domains involved (324 trap locations). The number of steps in Fish's inner loop $S$ cannot grow with the number of domains as this will make the gradient inner product approximation less valid (see details in Appendix A.2). We believe this is why Fish tends to underperform on these datasets with larger number of domains.
>
>
> > Is there any way one can check or compare - how well the linear approximation is working? If given infinite resources, can authors improve their scores further? If authors can show this on at least a few dataset, it will really help to understand the method better.
>
> Thank you for your suggestion. We do show the results of IDGM along with its approximation (Fish) in the linear example in section 2 and the color-dsprites example in section 3, where Fish's performance is very similar to IDGM. In section 4.4, we also show that Fish does maximise the gradient inner product while with ERM the gradient inner product goes to zero.
>
> We believe that if given infinite resources we would be able to improve our results by optimising IDGM directly, and this will be especially beneficial on datasets with larger number of domains as this is when Fish's approximation to IDGM becomes less accurate (as mentioned in the response to your previous question). As it stands, a model with IDGM objective trained on VLCS dataset with batch size 32 and standard ResNet18 backbone does not fit a 32GB GPU (NVIDIA V100), which is why we couldn't show results of IDGM for larger datasets.
>
>
> > How is the concept of using gradient inner product different from using kernel methods? Can authors compare and contrast with a popular DG method [1]?
>
> Our work learns invariant representation by maximising the gradient inner product, while [1] augments the original feature space with the marginal distribution of feature vectors and focuses on kernel methods, which is intrinsically different. Is there anything in particular that the reviewer want us to compare against?
>
> > Gradient inner products are also related to Neural Tangent Kernels [2]. Can authors comment more on this?
>
> Gradient inner product is used in [2] to linearly *approximate* infinitely wide nerual networks. In our work, it is *optimised* as part of the objective. Despite sharing the gradient inner product element, our two works are very different.

---

> > ### Comment · Reviewer_t4Z6 · 2021-11-26
> > **Reply**
> >
> >
> >
> > 1) "How is the concept of using gradient inner product different from using kernel methods? Can authors compare and contrast with a popular DG method [1]"
> > Authors should compare these kernel based DG methods in their paper. Even though there are differences, authors should include these papers in their literature survey.
> >
> > 2) "The number of steps in Fish's inner loop cannot grow with the number of domains as this will make the gradient inner product approximation less valid"  and
> > the statement from one of the other reviewer comments - "When the number of domains is large, the effect of gradient inner product maximisation using the Fish approximation diminishes (see Appendix A2 for more details). We believe this is why our method seem less competitive on datasets with high number of source domains.":
> >
> > Increase in number of domains and number of examples should increase domain generalization performance (unless there is some kind of negative transfer in the data). Authors say that approximation reduced the performance of their method. It is not clear to me if the exact method would definitely perform better in the case of large number of domains.
> >
> > [1] Blanchard, Gilles, Aniket Anand Deshmukh, Ürün Dogan, Gyemin Lee, and Clayton Scott. "Domain Generalization by Marginal Transfer Learning." arXiv preprint arXiv:1711.07910 (2017) J. Mach. Learn. Res. (JMLR) 22 (2021): 2-1.
> >
> > [2] Muandet, Krikamol, David Balduzzi, and Bernhard Schölkopf. "Domain generalization via invariant feature representation." In International Conference on Machine Learning, pp. 10-18. PMLR, 2013.
> >
> > [3] Erfani, Sarah, Mahsa Baktashmotlagh, Masud Moshtaghi, Xuan Nguyen, Christopher Leckie, James Bailey, and Rao Kotagiri. "Robust domain generalisation by enforcing distribution invariance." In Proceedings of the Twenty-Fifth International Joint Conference on Artificial Intelligence (IJCAI-16), pp. 1455-1461. AAAI Press, 2016.
> >
> > [4] Deshmukh, Aniket Anand, Yunwen Lei, Srinagesh Sharma, Urun Dogan, James W. Cutler, and Clayton Scott. "A generalization error bound for multi-class domain generalization." arXiv preprint arXiv:1905.10392 (2019).
> >
> > [5] Grubinger, Thomas, Adriana Birlutiu, Holger Schöner, Thomas Natschläger, and Tom Heskes. "Domain generalization based on transfer component analysis." In International Work-Conference on Artificial Neural Networks, pp. 325-334. Springer, Cham, 2015.
> >
> > I am still tending to accept the paper will keep my current score to 6.

---

### Official Review · Reviewer_Xwsw · 2021-11-03

**Correctness:** 3
**Technical Novelty And Significance:** 3
**Empirical Novelty And Significance:** 3
**Recommendation:** 6
**Confidence:** 3

**Main Review:**

Pros:

1. This paper proposes to maximize the similarity of gradients of the classifcation loss for different domains to learn invariant features. This is an interesting way to address the domain generalization problem.

2. This method also provides a effcient way to approximate the inner product between gradients due to the large computation cost. A theoretical proof is also given for this approximation.

Cons:

1. The improvment is minor on DomainBed in Table 3. The results of baseline methods seem to be inconsistent with the results by facebook?
https://github.com/facebookresearch/DomainBed/blob/main/domainbed/results/2020_10_06_7df6f06/results.png  Since this paper is claiming that it is trying to obtain domain-invariant features which is also what  DANN wants to achieve, then why DANN is worse than ERM while the IDGM is better than ERM? It would be better to have an explanation for this observation and full comparison with DANN.

2. It would be better to visualize (e.g., t-sne) the learned representations by the IDGM to see if the features are invariant across domains.


Update:

I have read the response by authors and other reviews. Part of my concern is addressed. So I raised my score to 6.
But I still share the same concern with another reviewer that it is unclear why this method works. If some domain-specific information is useful according to the response, then why the gradient matching helps preserves such information?

**Summary Of The Paper:**

This paper proposes to maximize the similarity of the gradients of the classification loss for different domains to learn domain-agnostic features.

**Summary Of The Review:**

This paper proposes to maximize the gradient similarity to learn invariant features across domains. But the improvment is minor and some critical explanations to the results are lacking.

---

> ### Author Response · Authors · 2021-11-15
> **Response to reviewer**
>
>
> > The results of baseline methods seem to be inconsistent with the results by facebook?  https://github.com/facebookresearch/DomainBed/blob/main/domainbed/results/2020_10_06_7df6f06/results.png
>
>
> Thank you for pointing out the inconsistency! For the baseline results of DomainBed we used the ones published with their paper (see [1], Table 4 "*training domain validation set*"). It does seem that the one listed by the reviewer is more up-to-date. We have updated this in our manuscript (see Table 4).
>
> Note that the updated results for all but one method (Coral) are **lower** than the reported results in [1] and in the previous version of our paper. Therefore this updates does not change our key finding -- Fish remains one of the best performing methods on DomainBed.
>
>
> > Since this paper is claiming that it is trying to obtain domain-invariant features which is also what DANN wants to achieve, then why DANN is worse than ERM while the IDGM is better than ERM? It would be better to have an explanation for this observation and full comparison with DANN.
>
>
>
> DANN imposes a harsh constraint on learning invariance, that is, all domain-specific information in the representation should be discarded, and the prediction of the label cannot depend on the domain at all. IDGM does not enforces such conditions, and instead only favours the learning of invariant directions by encouraging gradient alignment. Similarly, IRM also relax this constraint and learn invariant features by learning a universally optimal linear classifier.
>
> We would like to emphasise that the goal of many domain generalisation algorithms are to learn invariance, however they define invariance in different ways. For this reason their performances vary even though the motivations are similar.
>
>
>
>
> > It would be better to visualize (e.g., t-sne) the learned representations by the IDGM to see if the features are invariant across domains.
>
> Thank you for your suggestion! This is a great idea and we have included the t-SNE plot for VLCS, PACS and OfficeHome datasets in Appendix J of our updated manuscript. As we show in these plots, representations learned from both ERM and Fish form clear label-clusters for PACS and VLCS, however with ERM there are visible separation of domains within some of the clusters, whereas the same is not observed for Fish; On the other hand, for the OfficeHome dataset where Fish outperforms ERM by more than 1%, we clearly see that Fish exhibits better clustering performance than ERM.
>
>
> **Reference:**
>
> [1] I. Gulrajani and D. Lopez-Paz. In search of lost domain generalization. arXiv preprint
> arXiv:2007.01434, 2020.

---

### Official Review · Reviewer_b5oJ · 2021-11-03

**Correctness:** 4
**Technical Novelty And Significance:** 3
**Empirical Novelty And Significance:** 3
**Recommendation:** 8
**Confidence:** 4

**Main Review:**

The paper address an important problem in OOD generalization. The paper is well-written and the contributions are stated clearly. I have a few questions and concerns.

Positive Points:

1. The motivation and methodology is stated clearly. I can understand the approach by only read the paper once. I like figure 1 and section 3.2, which make the formulation itself very clear.

2. Despite the simplicity of Fish, it achieves non-trivial improvement on WILDS benchmark, outperforming important baselines such as IRM. It good to see some approaches finally outperform ERM by a large margin, e.g., CAMELYON17.

3. I like the extensive analysis in the experimental section, which provide valuable insights for future study.

Questions:

1. I think objective (4) shares similar motivation to IRM in the sense that we want the model to update in directions (use features) that are beneficial for all domains. It would be great to provide more discussion and comparison w.r.t. IRM.

2. I wonder whether the inner product of gradient in equation (4) could be related to bi-level / meta learning? For instance, check equation (12) in [1]. One may be able to explain the success of IDGM from the perspective of optimization, and further relates it to IRM. In particular, IRM defines its objective by measuring optimality, which is highly related to optimization. Note that this is different from analyzing approximation of IDGM such as Thm 3.1.

3. The measure drawback of the paper is it lacks of rigorous analysis about why and how IDGM works. For instance, IRM performs a careful study to analyze the proposed objective from a causal perspective. One may also investigate the problem from information theory [2]. It would be great if the paper can have at least a few paragraphs for analyzing IDGM.

Overall, there is no significant flaw in this paper. Although the proposed approach is not well analyzed, the empirical results are still impressive.

[1] Ren et al., Learning to Reweight Examples for Robust Deep Learning, ICML 2018

[2] Huszár, Invariant Risk Minimization: An Information Theoretic View, 2019

**Summary Of The Paper:**

To improve the generalization in unseen domains, the paper proposes a regularizer based on align ing the gradient direction between training environments. The proposed approach achieve non-trivial improvement on bench marks such as WILDS.

**Summary Of The Review:**

Overall, I think this is a well-written paper with good empirical results. I did not observe obvious flaw in the paper.

---

> ### Author Response · Authors · 2021-11-15
> **Response to reviewer**
>
>
> > I think objective (4) shares similar motivation to IRM in the sense that we want the model to update in directions (use features) that are beneficial for all domains. It would be great to provide more discussion and comparison w.r.t. IRM...one may be able to explain the success of IDGM from the perspective of optimization, and further relates it to IRM.
>
> Thank you for your suggestion! We have included a more detailed discussion comparing our method to IRM in our paper (see section 2). Please see below for our response to your questions:
>
>
>
>
> From an optimisation perspective, when IRM reaches its optimal, all the gradients **has to be zero**. This is why IRM's solution won't deviate from ERM when ERM is optimal for every domain -- as an example, when given the linear task we described in section 3.2, IRM will reach the same trivial solution as ERM (in other words, ERM's solution one of the valid IRM solutions). However, for IDGM, the gradients are not necessarily zero when the objective is optimised due to the gradient inner product term (this phenomena is shown empirically in Figure 5), and this is precisely why it is able to favour the invariant solution in our linear example, even at a cost of a higher loss for the ERM objective.
>
>
>
>
>
>
>
> > The measure drawback of the paper is it lacks of rigorous analysis about why and how IDGM works. For instance, IRM performs a careful study to analyze the proposed objective from a causal perspective. One may also investigate the problem from information theory [2]. It would be great if the paper can have at least a few paragraphs for analyzing IDGM.
>
> We hope the discussions above comparing against IRM from an optimisation perspective can serve as some insights on how IDGM works, which we will add to our paper.
>
> While it is true that our paper lack the theoretical analysis like in IRM, we do however have an intuitive explaination for our objective, a toy linear problem to observe its behaviour under linear setting, and interesting experimental results. We believe that these suffice as justification for our proposed method.
>
>
> > I wonder whether the inner product of gradient in equation (4) could be related to bi-level / meta learning? For instance, check equation (12) in [1].
>
> Thank you for sharing the paper, and it does indeed have very similar motivation to IDGM! We added a citation to this paper in Section 2.
>
> Eq.12 in [1] includes a gradient inner product (GIP) term between training and validation examples, and the authors noted that the training example that results in large GIP is similar to validation examples and therefore should be upweighted. Applying the same principle to domain generalisation, we could upweight examples in different training domains that result in a large GIP. This should help us focus on data from different domains that contains similar semantic information.
>
> From this perspective, IDGM follows similar intuition -- by maximising the GIP actively, it focuses on the invariant directions of prediction. This also circumvent the need of having to adopt the data curation strategy, which could introduce bias to the dataset as discussed in [1].
>
> **Reference:**
>
> [1] Slowik, Agnieszka, and Léon Bottou. “Algorithmic Bias and Data Bias: Understanding the Relation between Distributionally Robust Optimization and Data Curation.” ArXiv Preprint ArXiv:2106.09467, 2021.

---

> > ### Comment · Reviewer_b5oJ · 2021-11-25
> > **Response**
> >
> > Thank you for the clarifications. The updates in section 2 and A.1 look good. I will raise my score to 8.

---

### Official Review · Reviewer_KYNz · 2021-11-03

**Correctness:** 2
**Technical Novelty And Significance:** 2
**Empirical Novelty And Significance:** 2
**Recommendation:** 6
**Confidence:** 3

**Main Review:**

- The paper is well written and is easy to follow
- Multiple benchmarks are used to evaluate the method

Concerns:
- the claim that feature learnt are domain invariant is not really backed by a theoretical explanation or empirical. One way would be to show using tsne plot, that indeed features align as the gradient have same sign.
- Algorithm 1 formulation looks similar to the Mean Teacher[1] formulation where \tilde{\theta} plays the role of student and \theta as of teacher.
- Regarding just empirical results Fish seems competitive but it is not clear in what cases it the best. Is just the low number of source domain it shines? If that's the case, more explanation is needed to highlight the reason.
- Related Work section is needs to be improved, some works with which it is compared isn't discussed, for eg. Coral.
- Why the data augmentation based methods are left out in the comparisons?

[1] https://arxiv.org/abs/1703.01780

**Summary Of The Paper:**

The work tries to tackle the problem of domain generalisation in multi-source setting. The main claim of the paper is that by maximising inner product  between gradients from different domains leads to better learning of domain invariant features. The provide a meta-learning inspired algorithm Fish to approximate the second-order derivates. The results on several domain generalisation dataset is shown.

**Summary Of The Review:**

The paper presents gradient alignment to induce feature invariance. Several comparisons are presented but paper lacks explanation on the central idea, that why invariance is achieved. Also, some clarifications mentioned in the main review are need.

---

> ### Author Response · Authors · 2021-11-15
> **Response to reviewer**
>
> > the claim that feature learnt are domain invariant is not really backed by a theoretical explanation or empirical. One way would be to show using tsne plot, that indeed features align as the gradient have same sign.
>
> Thank you for your suggestion! This is a great idea and we have included the t-SNE plot for VLCS, PACS and OfficeHome datasets in Appendix J of our updated manuscript. As we show in these plots, representations learned from both ERM and Fish form clear label-clusters for PACS and VLCS, however with ERM there are visible separation of domains within some of the clusters, whereas the same is not observed for Fish; On the other hand, for the OfficeHome dataset where Fish outperforms ERM by more than 1%, we clearly see that Fish exhibits better clustering performance than ERM.
>
>
> > Algorithm 1 formulation looks similar to the Mean Teacher[1] formulation where \tilde{\theta} plays the role of student and \theta as of teacher.
>
> This is a great observation, thank you for pointing this out! We have added relavent citation in our manuscript (see bottom of Section 2).
>
> Fish is indeed similar to the Mean Teacher method, where a teacher model (equivalent to $\theta$ in Algorithm 1) is computed using a moving average of the student model (equivalent to $\tilde{\theta}$ in Algorithm 1). However with Fish, the "student model" is replaced by the "teacher model" at the beginning of each inner-loop, whereas with Mean Teacher the two models are kept separated.
>
>
>
>
>
> > Regarding just empirical results Fish seems competitive but it is not clear in what cases it the best. Is just the low number of source domain it shines? If that's the case, more explanation is needed to highlight the reason.
>
> When the number of domains $S$ is large, the effect of gradient inner product maximisation using the Fish approximation diminishes (see Appendix A2 for more details). We believe this is why our method seem less competitive on datasets with high number of source domains.
>
> In saying this, we would also like to point out that a lot of other domain generalisation algorithms (Group DRO, Coral) also fail to outperform ERM on these datasets. We are very interested in these failure modes of domain generalisation algorithms and are working towards addressing this phenomena.
>
> > Related Work section is needs to be improved, some works with which it is compared isn't discussed, for eg. Coral.
>
> We did cite and discussed Coral in our related work, however indeed we left out one work, MixUp, that we compared against via DomainBed in the related work section. We have revised section 2 of our manuscript based on this suggestion, thank you for pointing this out!
>
> > Why the data augmentation based methods are left out in the comparisons?
>
> The two benchmark we compared against include a total of 15 state-of-the-art methods, which we believe is sufficient to show the effectiveness of our model. We therefore did not implement additional methods in these benchmarks due to limited computational resources and time.

---

> > ### Comment · Reviewer_KYNz · 2021-11-26
> > **Response to Authors**
> >
> > Thank you for your clarifications and for highlighting the differences with previous methods in the updated manuscript. I will keep my rating the same.

---

### Author Response · Authors · 2021-11-15
**Revision uploaded**

We thank all the reviewers for their comments, and we have published the updated manuscript with the following changes:


1. **[KYNz, Xwsw]** Add t-SNE plot to show the invariance of learned representations (Appendix J)
2. **[KYNz]** Add citation for Mean Teacher formulation [1] (Section 2)
3. **[KYNz]** Discuss MixUp in related work (Section 2)
4. **[eig7]** Clarify that ERM doesn't completely fail in all domain generalisation tasks (Section 2)
5. **[b5oJ]** Add more thorough discussion about our work in comparison to IRM [3] (Section 2)
6. **[b5oJ]** Added citation to [4] (Section 2)
7. **[Xwsw]** Update DomainBed baseline results from the version seen in their [paper](https://arxiv.org/abs/2007.01434) to the most up-to-date online version as seen [here](https://github.com/facebookresearch/DomainBed/blob/main/domainbed/results/2020_10_06_7df6f06/results.png) (Table 4)
8. **[Da Li]** Added discussion about [5] (Section 2)

**References:**

[1] Tarvainen, Antti, and Harri Valpola. “Mean Teachers Are Better Role Models: Weight-Averaged Consistency Targets Improve Semi-Supervised Deep Learning Results.” ICLR (Workshop), 2017.

[2] Yan, Shen, et al. “Improve Unsupervised Domain Adaptation with Mixup Training.” ArXiv: Machine Learning, 2020.

[3] Arjovsky, Martin, et al. “Invariant Risk Minimization.” ArXiv Preprint ArXiv:1907.02893, 2019.

[4] Ren, Mengye, et al. “Learning to Reweight Examples for Robust Deep Learning.” International Conference on Machine Learning, 2018, pp. 4334–4343.

[5] Li, Da, et al. “Sequential Learning for Domain Generalization.” European Conference on Computer Vision, 2020, pp. 603–619.

---

### Public Comment · ~Da_Li3 · 2021-11-17
**Discussion of some related works?**

Thanks for the nice work on gradient matching for improving multi source domain generalization.
There is prior work in this area that introduced similar ideas. It would be great if you could discuss this in your paper and explain how your contribution advances it.
Specifically, inner product of gradients for DG was already discussed in this paper https://arxiv.org/pdf/1710.03463.pdf. (Compare Eq 8 in that reference vs Eq 4 in the current submission).
And there are also already followups using Reptile as a first-order approximation (https://arxiv.org/pdf/2004.01377.pdf,  Alg 2.)

---

> ### Author Response · Authors · 2021-11-20
> **Thank you for your comment!**
>
> We do indeed cite your work [1] in our paper, however we did miss your follow-up paper on this [2]. We will discuss the work in more details in our updated manuscript, thank you for pointing this out!
>
> Specifically, [2] also proposes to adapt Reptile for domain generlisation, however it is done under the sequential learning (by domain) setting, whereas our method can be trained on all domains and therefore learns faster, especially when the number of domains in the datasets are large.
>
> **References:**
>
> [1] Li, Da, et al. “Learning to Generalize: Meta-Learning for Domain Generalization.” Proceedings of the AAAI Conference on Artificial Intelligence, vol. 32, no. 1, 2018, pp. 3490–3497.
>
> [2] Li, Da, et al. “Sequential Learning for Domain Generalization.” European Conference on Computer Vision, 2020, pp. 603–619.

---

### Decision · Program_Chairs · 2022-01-20

**Decision:**

Accept (Poster)

**Comment:**

This paper proposes a new method for domain generalization. The main idea is to encourage higher inner-product between gradients from different domains. Instead of adding an explicit regularizer to encourage this, authors propose an optimization algorithm called Fish which implicitly encourages higher inner-product between gradients of different domains. Authors further show their proposed method is competitive on challenging benchmarks such as WILDS and DomainBed.

Reviewers all found the proposed algorithm novel and expressed that the contributions of the paper in terms of improving domain generalization is significant. A major issue that came up during the discussion period was that we realized that the presented results on WILDS benchmark are misleading. In particular, the following statements in the manuscript are false because on "CivilComments" and "Amazon", Fish utilizes a BERT model (Devlin et al., 2018). However, other methods at WILDS benchmark use DistilBERT (Sanh et al., 2019):

- Section 4.2: "For hyper-parameters including learning rate, batch size, choice of optimizer and model architecture, we follow the exact configuration as reported in the WILDS benchmark. Importantly, we also use the same model selection strategy used in WILDS to ensure a fair comparison."

- Appendix C2: "Results: We compare results to the baselines used in the WILDS benchmark over 3 random seed runs in Table 10. All models are trained using BERT (Devlin et al., 2018)."

Authors explained that the mismatch is because at the time they evaluated their model, an earlier version of WILDS benchmark was available but they later updated other methods' results on a newer version of WILDS benchmark. Of course, I do not think that this explanation makes the misleading statements OK. Authors promised to do the following for the camera-ready version to make sure it is not misleading:

- Using "Worst-U/R Pearson r" as the comparison measure for "PovertyMap"
- Submitting their method to WILDS benchmark making sure everything matches the baselines and then reporting the results on "Amazon" and "CivilComments" datasets.

Therefore, I recommend acceptance and I hope that authors would stick to their promise and update the manuscript to include these changes.